# Wake Interactions of Two Tandem Semisubmersible Floating Offshore Wind Turbines Based on FAST.Farm

Lei Xue 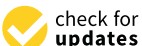, Jundong Wang, Liye Zhao, Zhiwen Wei, Mingqi Yu and Yu Xue *

College of Engineering, Ocean University of China, Qingdao 266100, China
* Correspondence: xueyu7231@ouc.edu.cn

**Abstract:** Wake effects commonly exist in offshore wind farms, which will cause a 10–20% reduction of whole power production as well as a 5–15% increase of fatigue loading on the wind turbine main structures. Obviously wake interaction between floating offshore wind turbine (FOWT) is more complicated, and needs careful assessment which is a prerequisite for active wake control (AWC). The primary objective of the present research is to investigate in detail how the wake inflow condition, streamwise spacing, turbulence intensity, and wind shear influence the power performance, platform motion dynamic and structural loading of FOWT. FAST.Farm, developed by the National Renewable Energy Laboratory (NREL), was used for simulating two tandem FOWTs in different conditions. Comparisons were made between FOWTs in different conditions on power performance and platform motion dynamic, which were presented through both time and frequency domain analysis. Damage equivalent loads change in FOWTs interference under typical working conditions were discussed and summarized. Half wake inflow would pose many challenges to the downstream FOWT. These research studies can be incorporated into further offshore wind farm wake models, providing applicable AWC strategies to reduce wake interference effects for higher energy production and for the longer life of FOWT.

**Keywords:** wind energy; floating wind turbine; wake effects; loads and response

## 1. Introduction

In order to achieve sustainable development and address global climate change, governments around the world are striving to develop wind power to provide clean energy. The 93.6 GW of new installations in 2021 brings global cumulative wind power capacity up to 837 GW by 2021. The offshore wind market has grown from 2.2 GW in 2016 to 21.1 GW in 2021, bringing its market share in global new installations from 4% to 22.5% [1]. It is estimated that floating offshore wind will grow from 17 MW in 2020 to 16.5 GW by 2030 [2]. It is foreseeable that offshore wind power will play an even more important role in the future of energy. However, wind power, especially offshore wind power, faces many urgent problems, such as the existence of the wake effect, reducing the power generation of downstream wind turbine (WT) and aggravating the fatigue load of WT structures. In large offshore wind farms, WT wakes increase fatigue loads on WT rotors by 5–15% while causing average power losses which are of the order of 10–20% of total power output [3–5].

Most of the research on wind farm wake is focused on wake loss models. The analysis model represented by the famous Jensen model [6,7] is one of the two kinds of wake loss model research methods. More comprehensive models, such as the Larsen model [8], Frandsen model [9], Gaussian models [10,11], and Geometric model [12], were proposed on this basis. On the other hand, the Eddy Viscosity Model (EVM) [13], the Deep-Array Wake Model (DAWM) [14], and the Large Array Wind Farm (LAWF) model [15] were computational fluid dynamics (CFD) -based models and were developed by making corrections to the original models, which were proved to be somewhat effective. The dynamic wake meander (DWM) model [16] was proven accurate at predicting single-turbine wake

development, providing a way to model both power production and loads on WTs in wind farms. Furthermore, another group of more sophisticated CFD-based models named Large-Eddy Simulation (LES) models have recently been used to study turbine wakes with great spatial and temporal resolution [17–21]. Nevertheless, LES models require large computational resources [22].

Some scholars have studied the influence of wind farm wake on WT load. Per Volund [23] concluded two main effects of wakes on the dynamic load of WT blades. In addition, wake-induced fatigue in offshore wind farm attracted attention [24,25]. Therefore, the influence of wake on the load of WT blades and other components in offshore wind farms is worth further study.

However, theoretical and numerical studies and experiments in the past mainly focused on the power loss caused by the wake effect. Some scholars have conducted numerical simulation research on wake interference between two WTs. A WT located downwind of another suffers a reduction in its aerodynamic power output if subject to impingement by the wake of the upstream WT. Simulations performed [26] suggest that WTs located even six times rotor diameter apart may experience a 40–50% reduction in power for tip speed ratios in the range six to eight. Computations [27] were carried out at different streamwise and crosswind displacements between the WTs, simulation study of flow field and power were done for both full wake and partial wake operation of the downstream WT. The wake interference between wind turbines in offshore wind farms is worth further study because of the high operation and maintenance costs.

Recently, there is a rapid demand for the development of floating wind farm because of the excellent wind energy resources in the abysmal sea [28], accompanied by the gradual deployment of spar and semisubmersible wind turbines [29]. Scholars [30] have found that despite nine rotor diameters between the turbines, wake effects were observed in the measured floater motions of the downstream floating offshore wind turbine (FOWT) during study on the Hywind Scotland wind farm which is known as the world's first fully operational floating wind farm.

The overview of existing literature reveals that there are few studies on wake interactions on FOWTs, especially for semi-submersible FOWTs. We hypothesized that certain wake inflow conditions could increase the risks of stability and reliability in semi-submersible FOWTs. A better understanding of the performance of FOWTs under different wake inflow condition may provide new insights into floating wind farm design, operation and maintenance and active wake control. As a result, the focus of this paper is on the following points:

(1)　The power performance, platform response and the load response of the semi-submersible FOWTs components were analyzed and evaluated under different wake inflow conditions.

(2)　Streamwise spacing, turbulence intensity (TI) and wind shear were considered in order to further study the influence of wake on the semi-submersible FOWTs.

The structure of this paper is laid out as follows. The software, model and simulation settings are introduced in Section 2. In Section 3, the power performance, platform response and structural loading under different conditions are compared and analyzed by statistical analysis in the time domain and power spectrum analysis in the frequency domain. Finally, Section 4 presents this study's conclusions about how different wake conditions affect performance in semi-submersible FOWTs, and discusses the shortcomings of this paper, considering the future research direction.

## 2. Methods and Materials

### 2.1. Dynamic Analysis Tool

FAST.Farm is a midfidelity multiphysics engineering tool [31] developed by the National Renewable Energy Laboratory (NREL) for predicting the power performance and structural loads of WTs within a wind farm. A summary of the simulation approach comparison is provided in Table 1. FAST.Farm considers additional physics for wind

farm-wide ambient wind in the atmospheric boundary layer, meanwhile, it uses OpenFAST to solve the aero-hydro-servo-elastic dynamics of each individual turbine [31]. FAST.Farm has one driver and four modules: Super Controller Module, OpenFAST Module, Wake Dynamics Module, Ambient Wind and Array Effects Module. In addition, TurbSim [32], a preprocesssor to FAST.Farm, was used for generating turbulent wind files.

**Table 1.** Comparison of simulation approach.

| Simulation Approach | Principle | Fidelity | Computational Expense | Structural Method |
|---|---|---|---|---|
| Semi empirical model | Wake loss model | Low-fidelity | Low | N/A |
| FAST.Farm | Dynamic wake meandering | Mid-fidelity | Low | ElastoDyn |
| SOWFA | Large eddy simulation | High-fidelity | High | Rigid body |

FAST.Farm is based on the principles of the DWM model which typically includes three submodels: velocity deficit, wake meandering and wake-added turbulence [31]. The information needed to calculate various parameters, as shown in Figure 1.

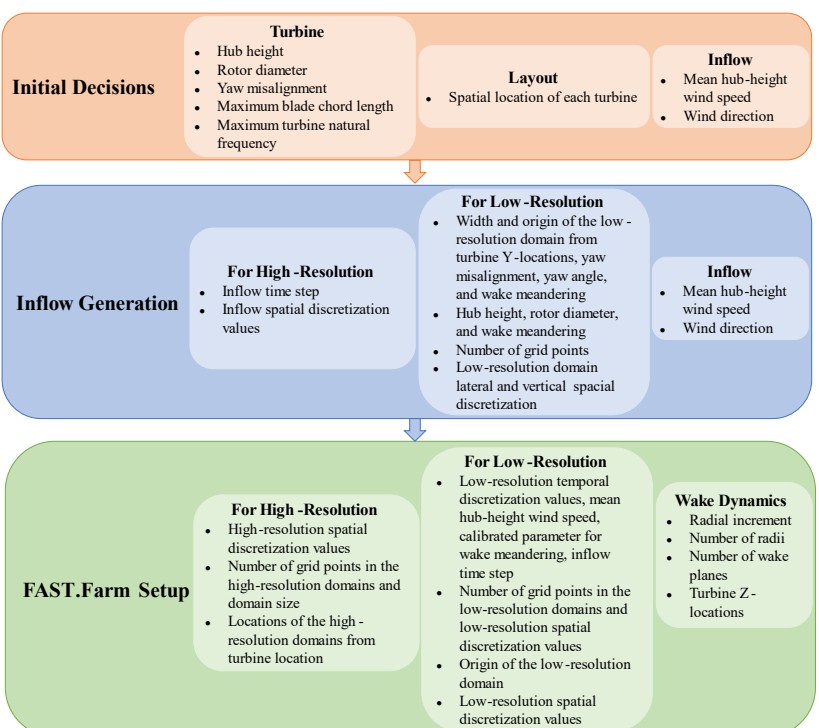

**Figure 1.** Information flowchart for FAST.Farm simulations [31].

## 2.2. Turbine Characteristics

In this study, all simulations concern the NREL 5 MW OC4-DeepCwind semi-submersible FOWT. The rated power of the FOWT is 5 MW, while the diameter of the rotor and hub height are 126 m and 90 m, respectively. Furthermore, cut-in and rated rotor speed respectively are 6.9 rpm and 12.1 rpm. Please refer to [33,34] for more detailed parameters of the FOWT.

## 2.3. Ambient Wind and Wave Conditions

Eighteen different wind files were generated by TurbSim for FAST.Farm, accompanied by 3 kinds of mean wind speed, 3 kinds of TI, and 2 kinds of wind shear. White noise spectrum was used for irregular wave model. Ambient wind and wave conditions were

shown in Table 2 in which wind parameter settings refer to an offshore wind farm located about 10 km from the coast of Jiangsu province, China [35,36].

**Table 2.** Wind conditions and wave condition setting.

| Parameter | Value | | |
|---|---|---|---|
| Mean wind speed | 8 m/s | 10 m/s | 12 m/s |
| TI | 0.06 | 0.08 | 0.10 |
| Turbulence model | | IECKAI | |
| Shear power law exponent | | 0.13 | 0.17 |
| Significant wave height of incident waves | | 1.2646 m | |
| Peak-spectral period of incident waves | | 10 s | |

In addition, The IEC Kaimal (IECKAI) model [32] assumes neutral atmospheric stability, which is defined in IEC 61400-1 3rd edition [37]. The spectra for the three wind components ($K = u, v, w$), are given by

$$S_K(f) = \frac{4\sigma^2 L_K/u}{(1 + 6f \cdot L_K/u)^{5/3}} \tag{1}$$

where $\sigma$ is the standard deviation and can be estimated by the TI, $f$ is the cyclic frequency, $L_K$ is an integral scale parameter and $u$ is the mean wind speed at hub height.

$$\sigma = TI \cdot u \tag{2}$$

where the integral scale parameter is defined to be

$$L_K = \begin{cases} 8.10 \cdot 0.7 \cdot 60, & K = u \\ 2.70 \cdot 0.7 \cdot 60, & K = v \\ 0.66 \cdot 0.7 \cdot 60, & K = w \end{cases} \tag{3}$$

The spectra of mean wind speed show that the low-frequency oscillations dominate the wind, which is displayed in Figure 2.

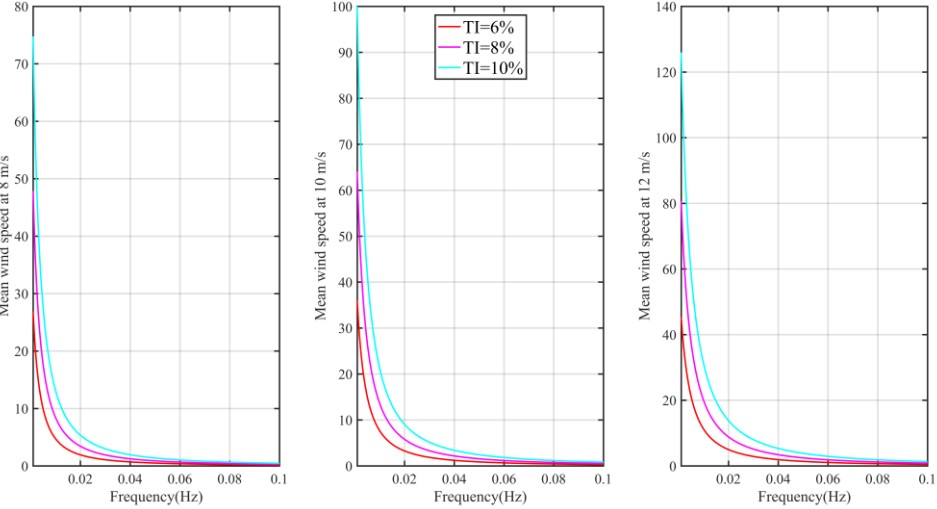

**Figure 2.** Spectrum of IECKAI turbulent wind.

*2.4. Simulation Settings*

For the simulation settings, a X × Y × Z = 6000 × 1000 × 350 m low-resolution domain and two X × Y × Z = 180 × 170 ×170 m high-resolution domains were used by FAST.Farm, and simulations were carried out with time length of 2000 seconds and

time step of 0.0125 second. The main flow direction was along X axis direction, and 6 km longitudinal length of the domain were sufficiently large enough for the wakes to propagate downstream, meanwhile, the lateral and vertical dimensions were large enough to allow for meandering of the wakes. Wind fields were clearly identifiable in Figure 3 [38].

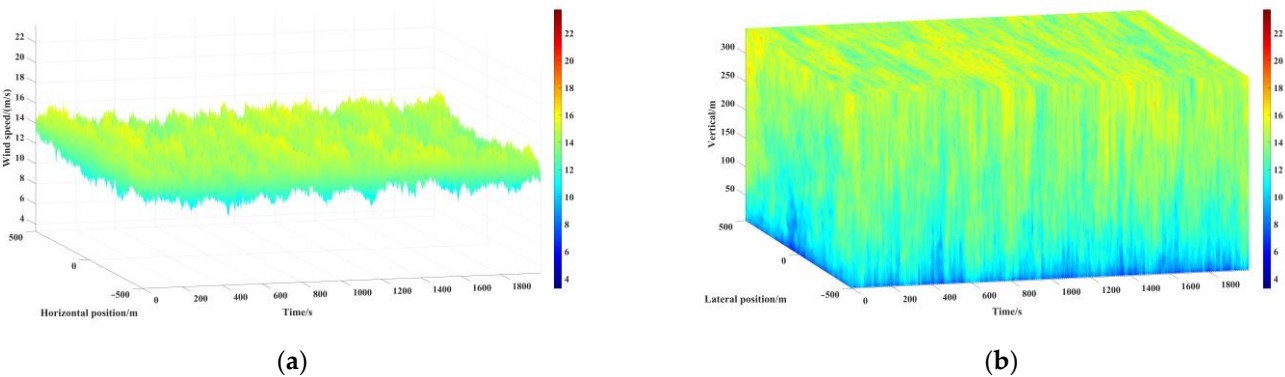

(**a**)                                                                                           (**b**)

**Figure 3.** The full-field turbulent wind with an average speed of 12 m/s: (**a**) time-varying wind speed at hub height; (**b**) full field wind speed distribution.

In lateral, the position of the FOWT 1 was fixed, while the position of the FOWT 2 varied at intervals of 10 m along the direction perpendicular to the main flow direction, as shown in Table 3. Numerous simulations were run within FAST.Farm on the operation of FOWTs under the conditions of 21 different kinds of wake conditions, such as no wake, quarter wake, half wake, three quarters wake and full wake inflow, which is illustrated in Figure 4a.

**Table 3.** FOWT 1 and FOWT 2 position setting.

| FOWT 1 (Xi, Yi) | FOWT 2 (Xi, Yi) (X = 630 m/1260 m) | | | | | |
|---|---|---|---|---|---|---|
| | (X, −200 m) | (X, −130 m) | (X, −60 m) | (X, 10 m) | (X, 80 m) | (X, 150 m) |
| | (X, −190 m) | (X, −120 m) | (X, −50 m) | (X, 20 m) | (X, 90 m) | (X, 160 m) |
| | (X, −180 m) | (X, −110 m) | (X, −40 m) | (X, 30 m) | (X, 100 m) | (X, 170 m) |
| (0 m,0 m) | (X, −170 m) | (X, −100 m) | (X, −30 m) | (X, 40 m) | (X, 110 m) | (X, 180 m) |
| | (X, −160 m) | (X, −90 m) | (X, −20 m) | (X, 50 m) | (X, 120 m) | (X, 190 m) |
| | (X, −150 m) | (X, −80 m) | (X, −10 m) | (X, 60 m) | (X, 130 m) | (X, 200 m) |
| | (X, −140 m) | (X, −70 m) | (X, 0 m) | (X, 70 m) | (X, 140 m) | |

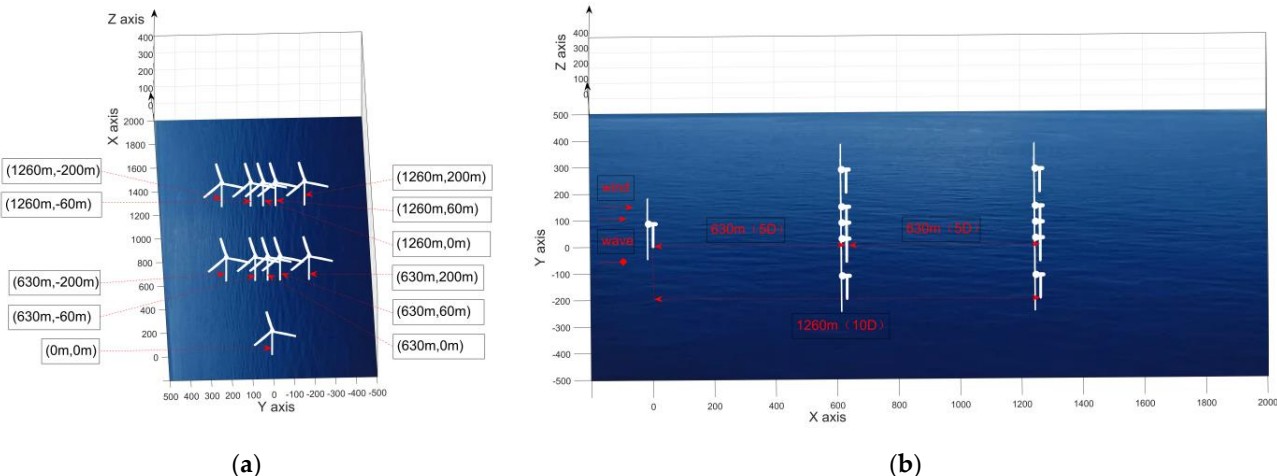

(**a**)                                                                                           (**b**)

**Figure 4.** The diagram of FOWT 1 and FOWT 2 representative position setting (ignore the platform): (**a**) longitudinal view; (**b**) lateral view.

In longitudinal, a simple two-WTs case where the downstream WT separated by 5D(630 m) and 10D(1260 m) in the wind direction was considered in order to explore the change of wake influence, as shown in Figure 4b.

## 3. Simulation Results

In this section, the performance of the downstream FOWTs were discussed in four subsections: time history exhibition, power performance, platform motion and structural loading, using the control variables.

### 3.1. Time History

Figure 5 shows a contour plot of wind speed at hub height with 5D streamwise spacing and 10D streamwise spacing, along with wind condition 12_0.06_0.13 (mean wind speed_ TI _ shear power law exponent).

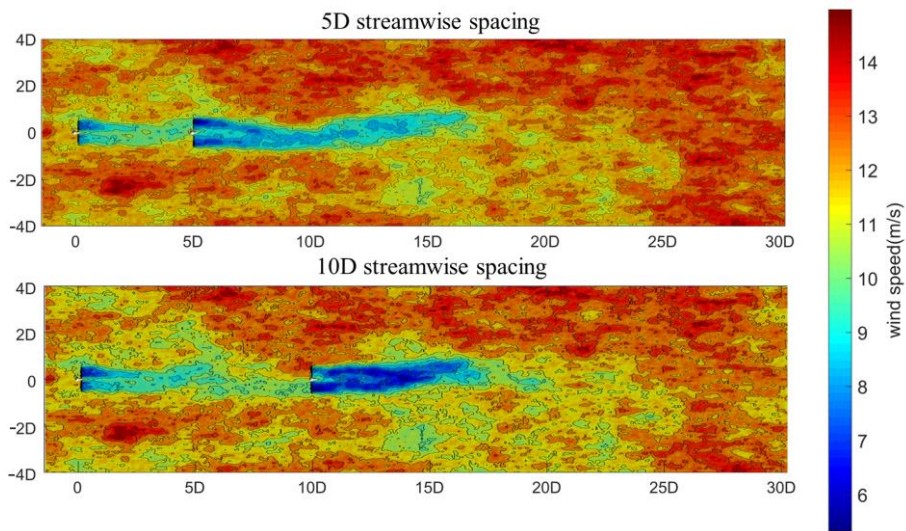

**Figure 5.** Contour plot of wind speed at hub height under full wake with 5D streamwise spacing and 10D streamwise spacing.

For all simulation cases in this study, the 600 s time series of 2000 s simulation was selected for analysis to avoid transient effects, which can be seen in Figure 6 taking power performance for example.

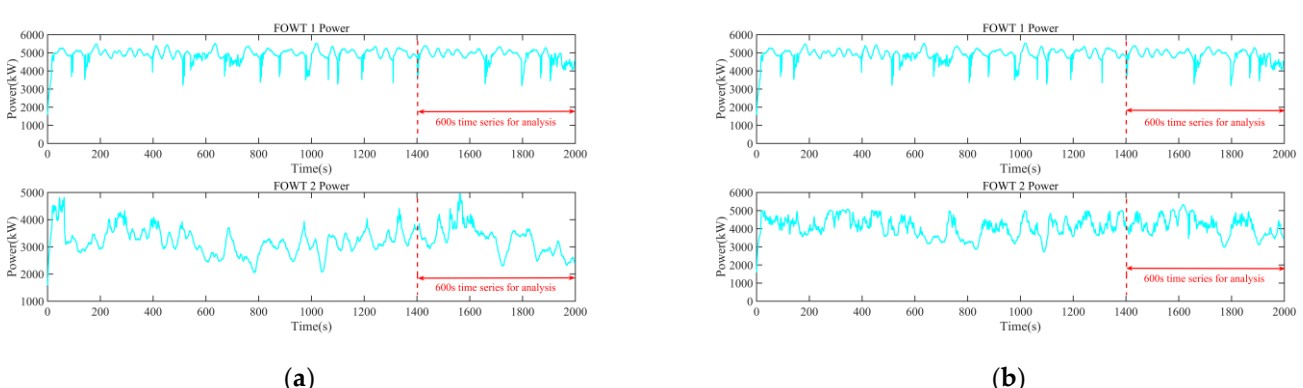

(**a**)                                                                                    (**b**)

**Figure 6.** Power time history of FOWT 1 and FOWT 2 on wind condition 12_0.06_0.13: (**a**) full wake, 5D streamwise spacing; (**b**) full wake, 10D streamwise spacing.

### 3.2. Power Performance

The power performance of the downstream FOWTs were discussed in five subsections, focusing on the FOWT 2 under different wake inflow condition, streamwise spacing, TI,

wind shear and power fluctuation. In Tables 4 and 5, the normalized processing took the rated power as the standard. As can be seen in Table 4, mean power loss of FOWT 2 under full wake condition can be around 20%, 35% and 25% under 5D streamwise spacing and mean wind speed at 8 m/s, 10 m/s and 12 m/s, respectively.

**Table 4.** Normalized mean power of FOWT 2 under full wake condition (standard: rated power).

| Case | Wind Condition | FOWT 1 | FOWT 2 (5D Streamwise Spacing) | FOWT 2 (10D Streamwise Spacing) | Interpolation (5D-10D Streamwise Spacing) |
|---|---|---|---|---|---|
| A1 | 8_0.06_0.13 | 0.35 | 0.15 (−0.20) | 0.23 (−0.12) | 0.08 |
| A2 | 8_0.08_0.13 | 0.35 | 0.16 (−0.19) | 0.24 (−0.11) | 0.08 |
| A3 | 8_0.10_0.13 | 0.35 | 0.17 (−0.18) | 0.26 (−0.09) | 0.09 |
| A4 | 8_0.06_0.17 | 0.35 | 0.15 (−0.20) | 0.23 (−0.12) | 0.08 |
| A5 | 8_0.08_0.17 | 0.35 | 0.16 (−0.19) | 0.25 (−0.10) | 0.08 |
| A6 | 8_0.10_0.17 | 0.35 | 0.18 (−0.17) | 0.26 (−0.09) | 0.09 |
| B1 | 10_0.06_0.13 | 0.66 | 0.30 (−0.36) | 0.46 (−0.20) | 0.16 |
| B2 | 10_0.08_0.13 | 0.67 | 0.32 (−0.35) | 0.48 (−0.19) | 0.16 |
| B3 | 10_0.10_0.13 | 0.67 | 0.35 (−0.32) | 0.51 (−0.16) | 0.16 |
| B4 | 10_0.06_0.17 | 0.66 | 0.30 (−0.36) | 0.46 (−0.20) | 0.16 |
| B5 | 10_0.08_0.17 | 0.67 | 0.33 (−0.34) | 0.49 (−0.18) | 0.16 |
| B6 | 10_0.10_0.17 | 0.67 | 0.35 (−0.32) | 0.51 (−0.16) | 0.16 |
| C1 | 12_0.06_0.13 | 0.97 | 0.66 (−0.31) | 0.84 (−0.13) | 0.18 |
| C2 | 12_0.08_0.13 | 0.95 | 0.69 (−0.26) | 0.86 (−0.09) | 0.17 |
| C3 | 12_0.10_0.13 | 0.94 | 0.73 (−0.21) | 0.88 (−0.06) | 0.16 |
| C4 | 12_0.06_0.17 | 0.97 | 0.67 (−0.30) | 0.84 (−0.13) | 0.18 |
| C5 | 12_0.08_0.17 | 0.95 | 0.70 (−0.25) | 0.87 (−0.08) | 0.17 |
| C6 | 12_0.10_0.17 | 0.94 | 0.73 (−0.21) | 0.88 (−0.06) | 0.15 |

**Table 5.** Normalized mean power of FOWT 2 with different position setting (standard: rated power).

| Case | FOWT 2 (5D Streamwise Spacing) | FOWT 2 (10D Streamwise Spacing) | Case | FOWT 2 (5D Streamwise Spacing) | FOWT 2 (10D Streamwise Spacing) | Case | FOWT 2 (5D Streamwise Spacing) | FOWT 2 (10D Streamwise Spacing) |
|---|---|---|---|---|---|---|---|---|
| A1_P1 | 0.35 | 0.36 | B1_P1 | 0.67 | 0.68 | C1_P1 | 0.97 | 0.98 |
| A1_P2 | 0.35 | 0.36 | B1_P2 | 0.67 | 0.67 | C1_P2 | 0.97 | 0.98 |
| A1_P3 | 0.35 | 0.35 | B1_P3 | 0.66 | 0.67 | C1_P3 | 0.97 | 0.98 |
| A1_P4 | 0.35 | 0.35 | B1_P4 | 0.66 | 0.66 | C1_P4 | 0.97 | 0.97 |
| A1_P5 | 0.34 | 0.34 | B1_P5 | 0.65 | 0.65 | C1_P5 | 0.96 | 0.97 |
| A1_P6 | 0.34 | 0.34 | B1_P6 | 0.64 | 0.64 | C1_P6 | 0.96 | 0.97 |
| A1_P7 | 0.33 | 0.33 | B1_P7 | 0.63 | 0.63 | C1_P7 | 0.96 | 0.96 |
| A1_P8 | 0.32 | 0.32 | B1_P8 | 0.61 | 0.62 | C1_P8 | 0.95 | 0.95 |
| A1_P9 | 0.31 | 0.32 | B1_P9 | 0.60 | 0.61 | C1_P9 | 0.94 | 0.94 |
| A1_P10 | 0.30 | 0.31 | B1_P10 | 0.57 | 0.59 | C1_P10 | 0.93 | 0.93 |
| A1_P11 | 0.28 | 0.30 | B1_P11 | 0.54 | 0.58 | C1_P11 | 0.91 | 0.92 |
| A1_P12 | 0.27 | 0.29 | B1_P12 | 0.51 | 0.56 | C1_P12 | 0.89 | 0.91 |
| A1_P13 | 0.25 | 0.28 | B1_P13 | 0.48 | 0.54 | C1_P13 | 0.86 | 0.90 |
| A1_P14 | 0.23 | 0.27 | B1_P14 | 0.45 | 0.53 | C1_P14 | 0.83 | 0.88 |
| A1_P15 | 0.21 | 0.26 | B1_P15 | 0.41 | 0.51 | C1_P15 | 0.79 | 0.87 |
| A1_P16 | 0.19 | 0.25 | B1_P16 | 0.38 | 0.50 | C1_P16 | 0.76 | 0.86 |
| A1_P17 | 0.18 | 0.24 | B1_P17 | 0.35 | 0.48 | C1_P17 | 0.72 | 0.85 |
| A1_P18 | 0.16 | 0.24 | B1_P18 | 0.33 | 0.47 | C1_P18 | 0.69 | 0.84 |
| A1_P19 | 0.15 | 0.23 | B1_P19 | 0.31 | 0.47 | C1_P19 | 0.67 | 0.84 |
| A1_P20 | 0.15 | 0.23 | B1_P20 | 0.30 | 0.46 | C1_P20 | 0.66 | 0.84 |
| A1_P21 | 0.15 | 0.23 | B1_P21 | 0.30 | 0.46 | C1_P21 | 0.66 | 0.84 |
| A1_P22 | 0.15 | 0.23 | B1_P22 | 0.31 | 0.46 | C1_P22 | 0.66 | 0.84 |
| A1_P23 | 0.16 | 0.23 | B1_P23 | 0.32 | 0.46 | C1_P23 | 0.68 | 0.85 |

**Table 5.** *Cont.*

| Case | FOWT 2 (5D Streamwise Spacing) | FOWT 2 (10D Streamwise Spacing) | Case | FOWT 2 (5D Streamwise Spacing) | FOWT 2 (10D Streamwise Spacing) | Case | FOWT 2 (5D Streamwise Spacing) | FOWT 2 (10D Streamwise Spacing) |
|---|---|---|---|---|---|---|---|---|
| A1_P24 | 0.17 | 0.24 | B1_P24 | 0.34 | 0.47 | C1_P24 | 0.70 | 0.86 |
| A1_P25 | 0.18 | 0.24 | B1_P25 | 0.36 | 0.48 | C1_P25 | 0.73 | 0.87 |
| A1_P26 | 0.20 | 0.25 | B1_P26 | 0.39 | 0.49 | C1_P26 | 0.77 | 0.88 |
| A1_P27 | 0.21 | 0.26 | B1_P27 | 0.42 | 0.51 | C1_P27 | 0.81 | 0.89 |
| A1_P28 | 0.23 | 0.27 | B1_P28 | 0.45 | 0.52 | C1_P28 | 0.84 | 0.91 |
| A1_P29 | 0.25 | 0.28 | B1_P29 | 0.48 | 0.54 | C1_P29 | 0.88 | 0.92 |
| A1_P30 | 0.26 | 0.29 | B1_P30 | 0.51 | 0.56 | C1_P30 | 0.90 | 0.93 |
| A1_P31 | 0.28 | 0.30 | B1_P31 | 0.54 | 0.57 | C1_P31 | 0.92 | 0.94 |
| A1_P32 | 0.29 | 0.30 | B1_P32 | 0.56 | 0.58 | C1_P32 | 0.94 | 0.95 |
| A1_P33 | 0.30 | 0.31 | B1_P33 | 0.58 | 0.60 | C1_P33 | 0.95 | 0.95 |
| A1_P34 | 0.31 | 0.32 | B1_P34 | 0.59 | 0.61 | C1_P34 | 0.96 | 0.96 |
| A1_P35 | 0.32 | 0.33 | B1_P35 | 0.61 | 0.62 | C1_P35 | 0.96 | 0.96 |
| A1_P36 | 0.32 | 0.33 | B1_P36 | 0.61 | 0.63 | C1_P36 | 0.97 | 0.97 |
| A1_P37 | 0.33 | 0.34 | B1_P37 | 0.62 | 0.64 | C1_P37 | 0.97 | 0.97 |
| A1_P38 | 0.33 | 0.34 | B1_P38 | 0.62 | 0.64 | C1_P38 | 0.97 | 0.97 |
| A1_P39 | 0.33 | 0.34 | B1_P39 | 0.63 | 0.65 | C1_P39 | 0.97 | 0.97 |
| A1_P40 | 0.33 | 0.34 | B1_P40 | 0.63 | 0.65 | C1_P40 | 0.97 | 0.97 |
| A1_P41 | 0.33 | 0.35 | B1_P41 | 0.63 | 0.66 | C1_P41 | 0.96 | 0.97 |

### 3.2.1. Wake Inflow Condition

Figures 7–9 and Table 5 showed discrepancy on the power performance between FOWTs under different wake inflow conditions. The power increased with almost the same trend from full wake to no wake at both ends. However, with the full wake as the center, the power performance on both sides was not completely symmetrical. It should be emphasized that FOWT 2s (A1_P1- A1_P20, B1_P1- B1_P20, C1_P1- C1_P20) whose rotor under right-half wake (Yi = −60 m) did not perform as well as FOWT 2s(A1_P22- A1_P41, B1_P22- B1_P41, C1_P22- C1_P41) whose rotor under left-half wake (Yi = 60 m), which can be seen in Table 5. Figure 9b illustrated that the difference between wake inflow conditions is no longer prominent under the action of sufficient streamwise spacing and sufficient TI and wind shear.

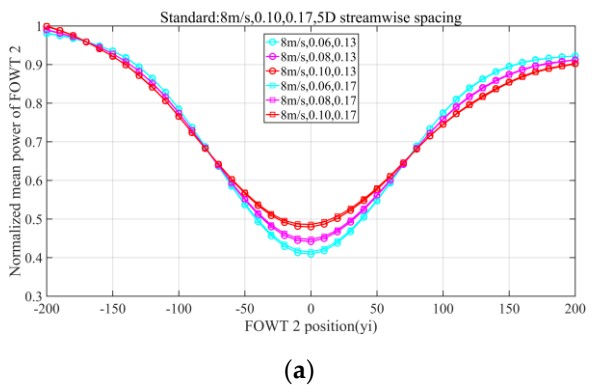

(**a**)

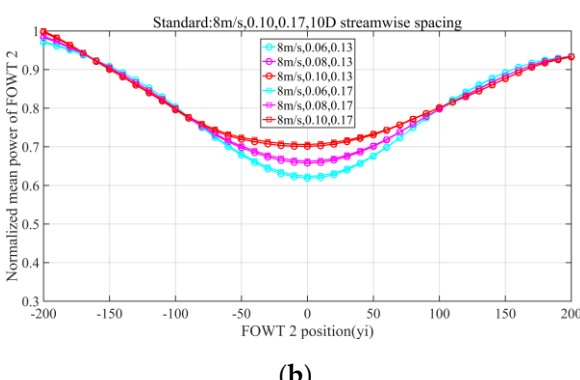

(**b**)

**Figure 7.** Normalized mean power of FOWT 2 under different wake inflow conditions: (**a**) 8 m/s, 5D streamwise spacing; (**b**) 8 m/s, 10D streamwise spacing.

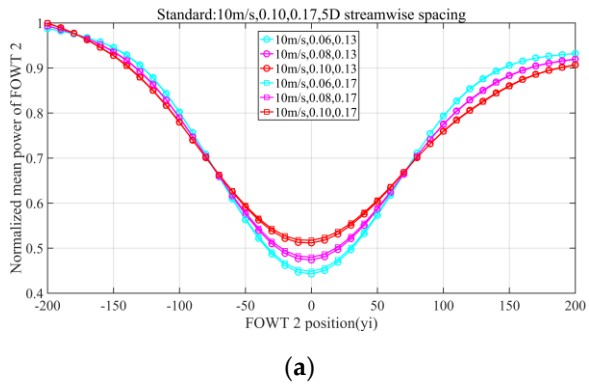
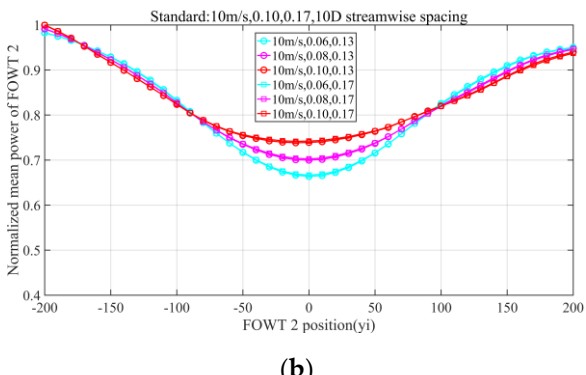

**Figure 8.** Normalized mean power of FOWT 2 under different wake inflow conditions: (**a**) 10 m/s, 5D streamwise spacing; (**b**) 10 m/s, 10D streamwise spacing.

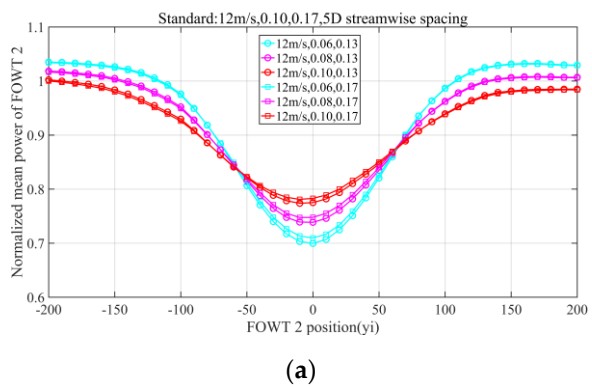
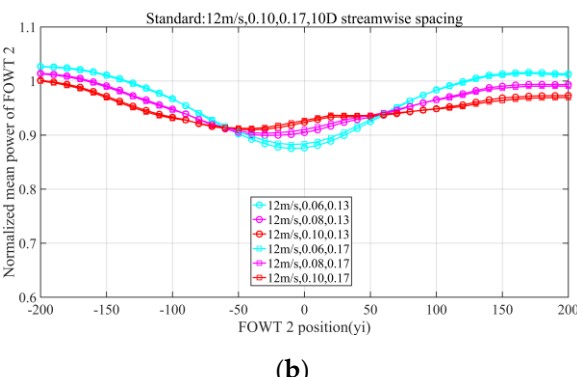

**Figure 9.** Normalized mean power of FOWT 2 under different wake inflow conditions: (**a**) 12 m/s, 5D streamwise spacing; (**b**) 12 m/s, 10D streamwise spacing.

### 3.2.2. Streamwise Spacing

Table 5 showed that increasing the distance effectively improves the power recovery, especially when it was close to the rated wind speed, which can also be seen from Figures 7a, 8a and 9a to Figures 7b, 8b and 9b. In addition, for the same wake inflow state, there was difference between the power recovery degree of FOWT 2 under 5D and 10D streamwise spacing. Increasing the distance can effectively mitigates wake effects, averagely increased FOWT 2 power 8%, 16% and 17% of rated power under wind speed at 8 m/s, 10 m/s and 12 m/s, respectively.

### 3.2.3. Turbulence Intensity

TI played an important role in power recovery of FOWT 2 under different wake conditions, as shown in Figures 7–9 which indicated that higher TI was beneficial to wake recovery of FOWT 2 s, together with Table 5. In addition, higher TI increased FOWT 2 power 1% of rated power under wind speed at 8 m/s approximately, as described in Table 5.

### 3.2.4. Wind Shear

Wind shear played a small role in mitigating wake influence at low wind speed, but played a more obvious role at close to rated wind speed. A power increase due to wind shear was not apparent with the exception of the circumstances when FOWT 2 under full wake, as shown in Table 5. The contribution of wind shear to wake recovery is more obvious at above rated wind speed than below rated wind speed, regardless of the streamwise spacing. Figures 7a, 8a and 9a showed distinctly that the effect of wind shear on wake recovery was less than that of TI.

### 3.2.5. Power Fluctuation

The wake not only caused the power loss of FOWT 2, but also increased the power fluctuation significantly, especially when rotor of FOWT 2 under right-half wake, as seen in Figure 10. Furthermore, this also indicated that in the actual operation of offshore wind farms, the occurrence of the phenomenon often brings great challenges to the wind power forecasting or battery energy storage systems [39], resulting in the change in active power of wind farms not being able to meet the requirements of safe and stable operation of power systems [40].

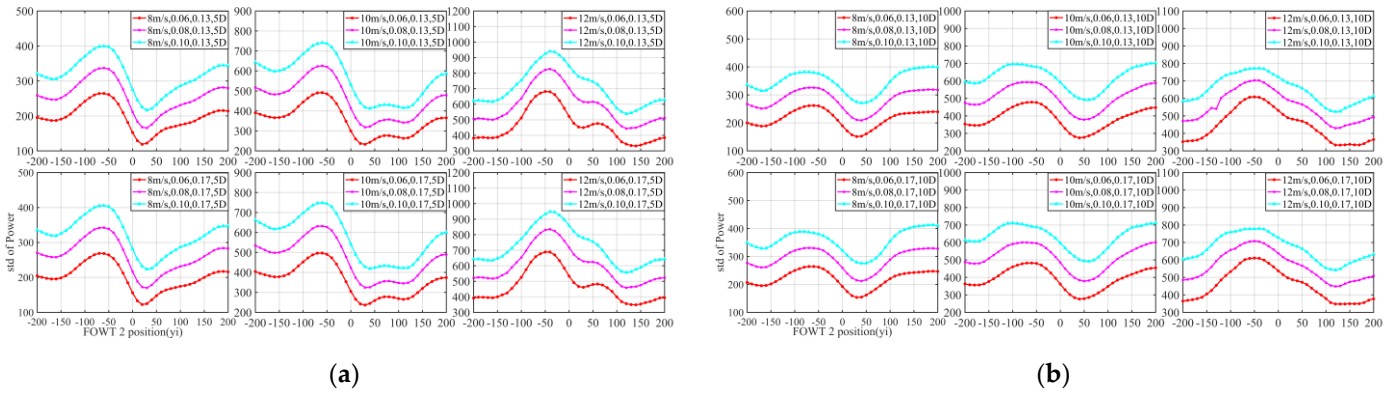

(**a**)　　　　　　　　　　　　　　　　　　　　　　　　　　(**b**)

**Figure 10.** STD of FOWT 2 power under different wake inflow conditions: (**a**) 5D streamwise spacing; (**b**) 10D streamwise spacing.

### 3.3. Platform Motion

The wake, as well as the TI and wind shear, may induce a complex response on FOWTs. Therefore, it is essential to investigate the platform motion response of the FOWT under severe states.

#### 3.3.1. Comparison of Time-Domain Response

Having indicated in Figures 11–16 the degrees of freedom (DoFs) surge, sway, heave, roll, pitch, and yaw motion statistics of FOWT 2 under different wake inflow conditions, including the maximum, minimum, average, and standard deviation (STD) values.

Different wake inflow conditions caused inequable fluctuations on platform DoFs, especially for surge and pitch. As observed in Figure 14a, the STD of surge exceed 20% while STD of surge exceed 30%, showing that the instability of the FOWT 2 platform increased as a result of half wake condition. After reaching rated speed, the average wind speed had more influence on the platform DoFs than the wake inflow conditions. In addition, with the full wake as the center, the platform DoFs performance on both sides is not completely symmetric as well.

Figures 11–16 showed that higher TI have evident impact on platform DoFs, exclusive of heave. STD of surge, sway, roll, pitch, and yaw motion were enlarged by 8–20%, 3–20%, 10–25%, 15–22% and 6–28%, as shown in Figure 14a.

In general, there were no significant effects on platform DoFs with variations in streamwise spacing as well as wind shear under different wake inflow conditions.

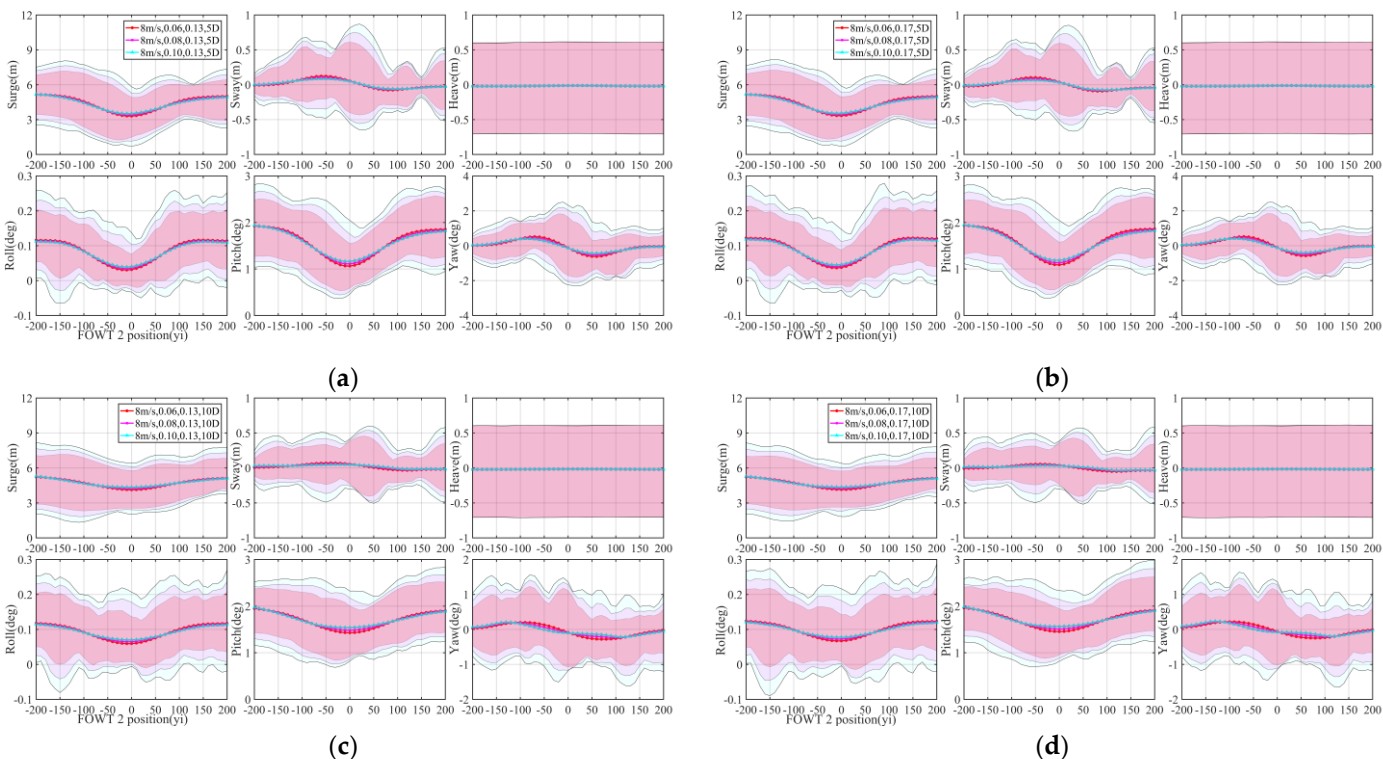

**Figure 11.** Maximum, minimum, and average of FOWT 2 platform motion under different wake inflow conditions: (**a**) 8 m/s, 5D streamwise spacing; (**b**) 8 m/s, 5D streamwise spacing; (**c**) 8 m/s, 10D streamwise spacing; (**d**) 8 m/s, 10D streamwise spacing.

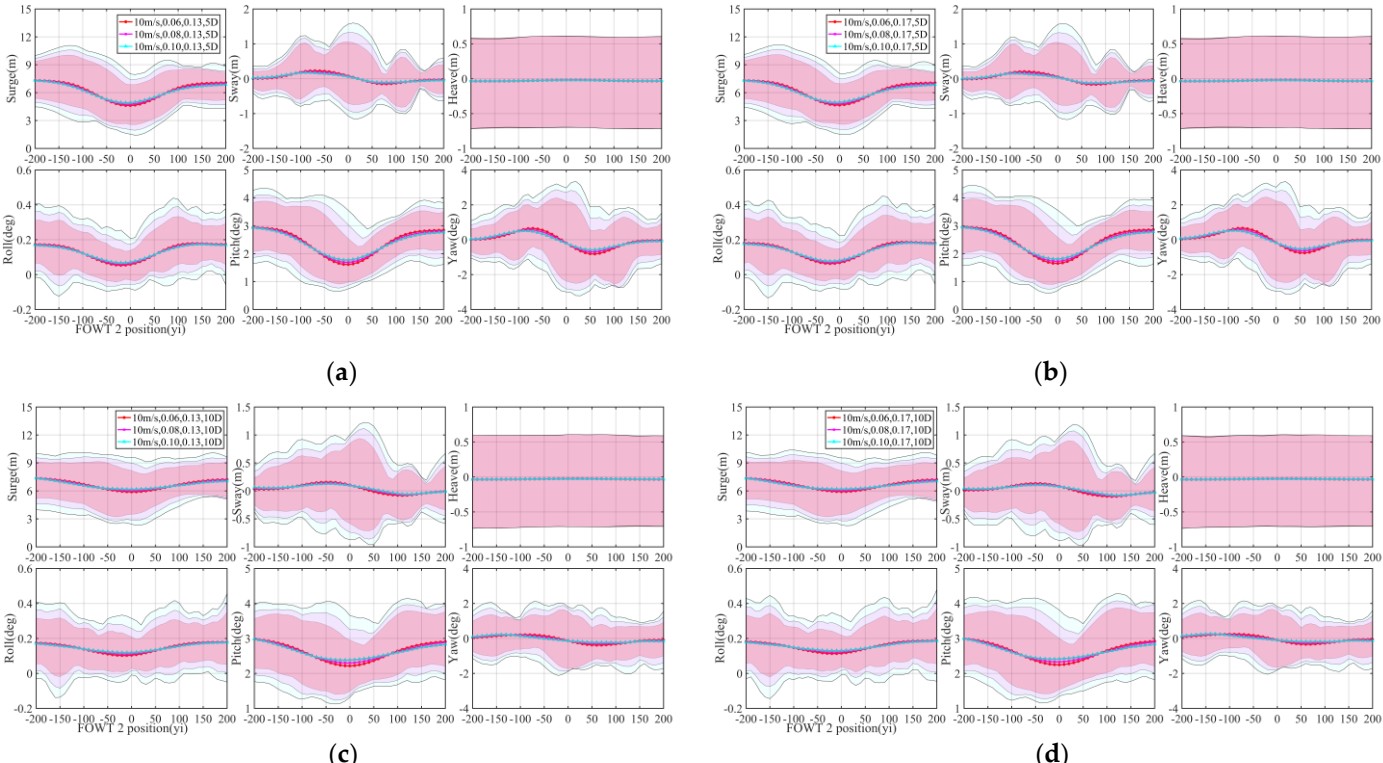

**Figure 12.** Maximum, minimum, and average of FOWT 2 platform motion under different wake inflow conditions: (**a**) 10 m/s, 5D streamwise spacing; (**b**) 10 m/s, 5D streamwise spacing; (**c**) 10 m/s, 10D streamwise spacing; (**d**) 10 m/s, 10D streamwise spacing.

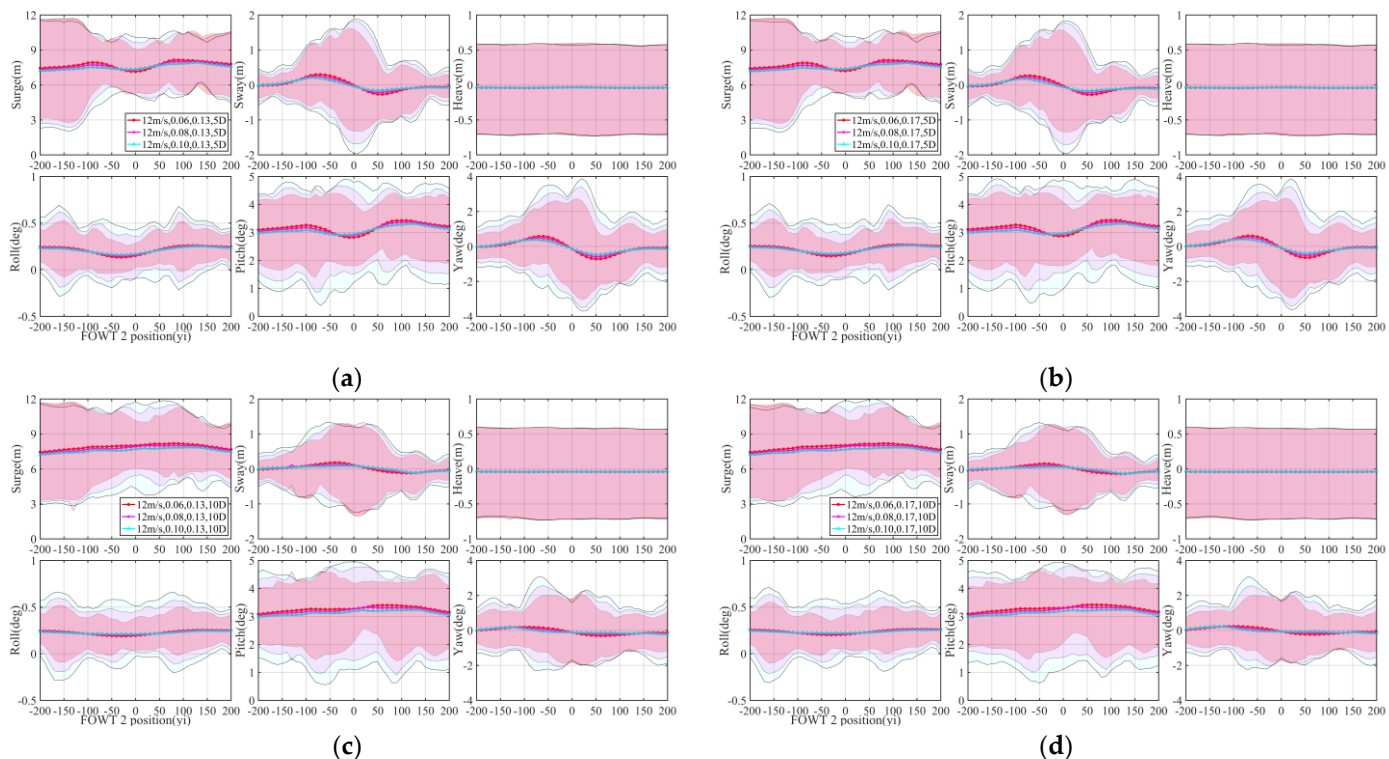

**Figure 13.** Maximum, minimum, and average of FOWT 2 platform motion under different wake inflow conditions: (**a**) 12 m/s, 5D streamwise spacing; (**b**) 12 m/s, 5D streamwise spacing; (**c**) 12 m/s, 10D streamwise spacing; (**d**) 12 m/s, 10D streamwise spacing.

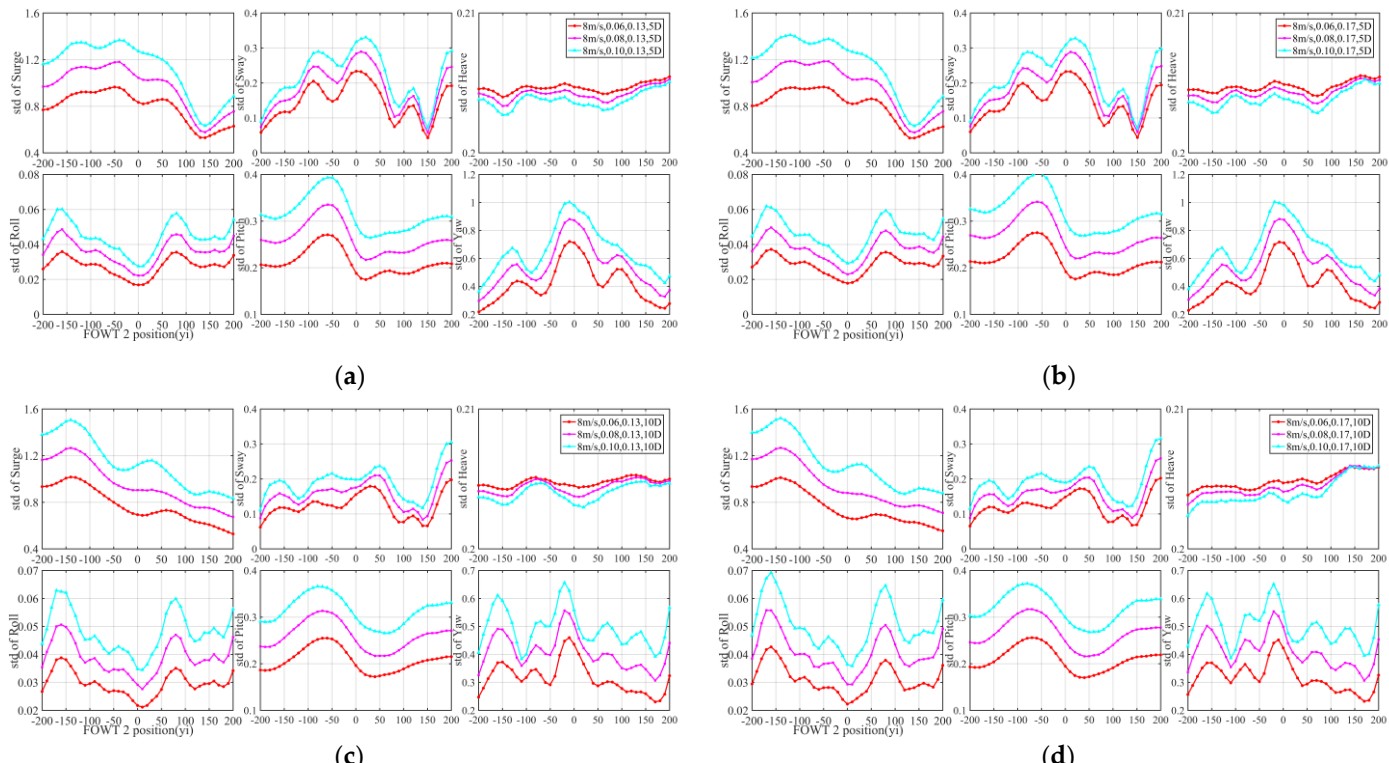

**Figure 14.** STD of FOWT 2 platform motion under different wake inflow conditions: (**a**) 8 m/s, 5D streamwise spacing; (**b**) 8 m/s, 5D streamwise spacing; (**c**) 8 m/s, 10D streamwise spacing; (**d**) 8 m/s, 10D streamwise spacing.

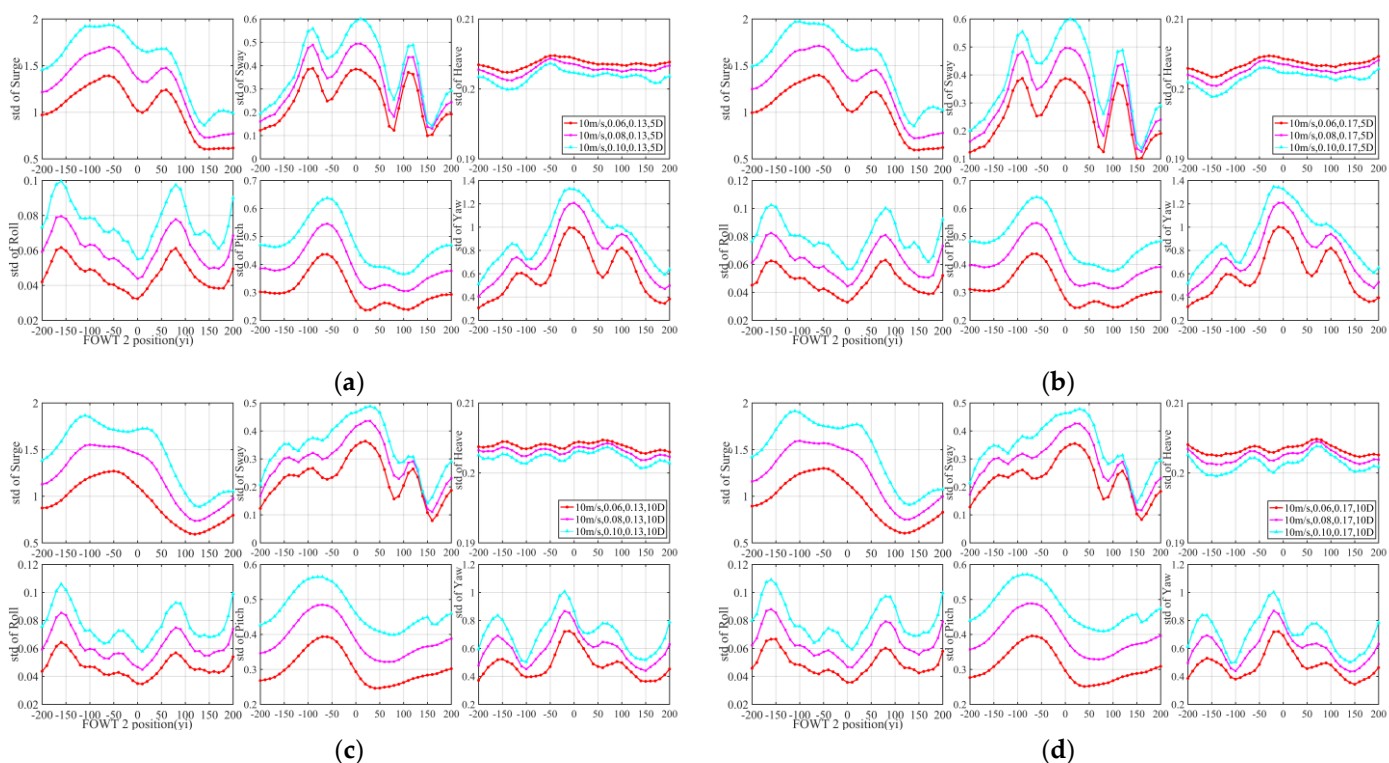

**Figure 15.** STD of FOWT 2 platform motion under different wake inflow conditions: (**a**) 10 m/s, 5D streamwise spacing; (**b**) 10 m/s, 5D streamwise spacing; (**c**) 10 m/s, 10D streamwise spacing; (**d**) 10 m/s, 10D streamwise spacing.

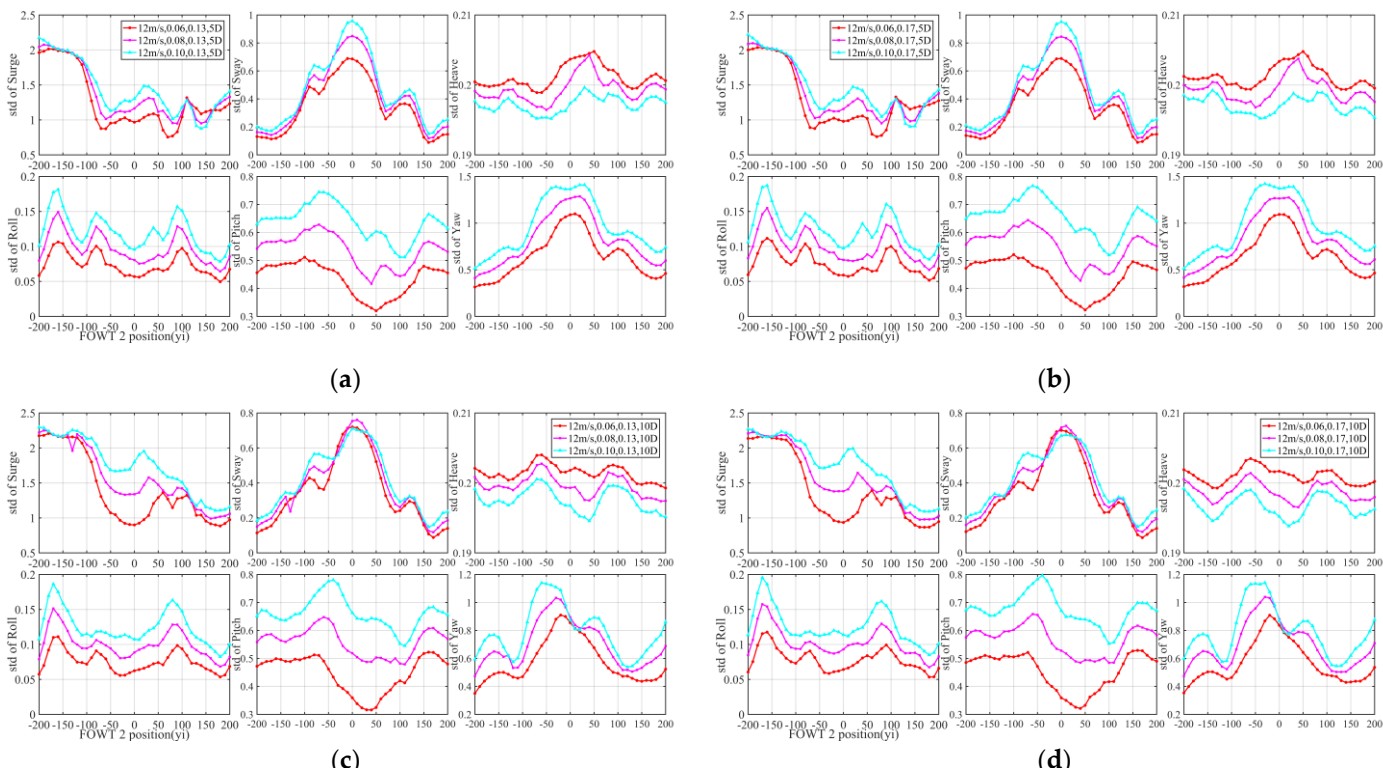

**Figure 16.** STD of FOWT 2 platform motion under different wake inflow conditions: (**a**) 12 m/s, 5D streamwise spacing; (**b**) 12 m/s, 5D streamwise spacing; (**c**) 12 m/s, 10D streamwise spacing; (**d**) 12 m/s, 10D streamwise spacing.

### 3.3.2. Power Spectrum Analysis

In this subsection, five typical conditions were selected for PSD calculation and more detailed insights on platform motion under different wake inflow conditions.

In contrast to the normal distribution of FOWT 1 hub center wind speed spectrum, spectral peaks of FOWT 2 hub center wind speed appeared significantly due to wake, especially in the case of half wake, as seen in Figure 17a. Figure 17b showed that peaks were mainly in the frequency band of 0.05–0.25 Hz at the selected wave condition. Since the FOWT 2 s were subject to identical wave excitations in the turbulent wind, the amplitude of platform motions could be purely attributed to the wind force.

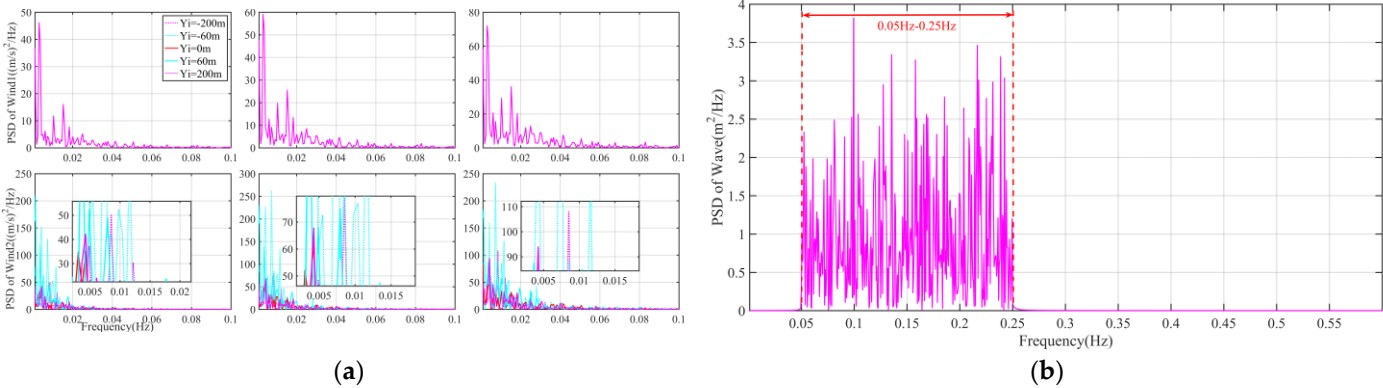

(**a**)                                                                                          (**b**)

**Figure 17.** PSD of hub center wind speed (FOWT 1 and FOWT 2) and PSD of wave elevation: (**a**) 8_0.06_0.13&10_0.06_0.13&12_0.06_0.13, 5D streamwise spacing; (**b**) wave elevation (global Z height).

The performance of platform motion under the different wake inflow conditions also confirmed the initial hypothesis. Resonant response at the surge natural frequency was greatest for the rated scenario, when the FOWT operated with the greatest thrust, meanwhile pitch responses followed a similar trend, this phenomenon was in line with [41], which were presented in Figures 18–20. Additionally, the unsteady low-frequency change of wind generation system, as well as the surge natural frequency, existed in the response spectra for pitch. Figure 20 showed that the frequency range of platform pitch responses at above rated wind speed is narrower than that at below rated wind speed.

It is interesting to note that the spectral response of surge and pitch was significantly enhanced when FOWT 2 encountered right-half wake rather than left-half wake or full wake, while the same trend was also reflected in power performance. Comparing the spectral data of other motions, it can be observed that full wake has more impact on the sway, heave, and yaw motion while wake reduced the response of the roll motion.

The platform motions performances of the FOWT 2 with different TI were shown in Figures 18a–e, 19a–e and 20a–e. The TI effect was only observed within the low-frequency region, and the platform motions became increasingly violent when the TI increased; a similar phenomenon had been documented previously in [42]. Furthermore, the influence of wind shear on the platform motion is insignificant compared with the change of TI, as illustrated in Figures 18a,b, 19a,b and 20a,b.

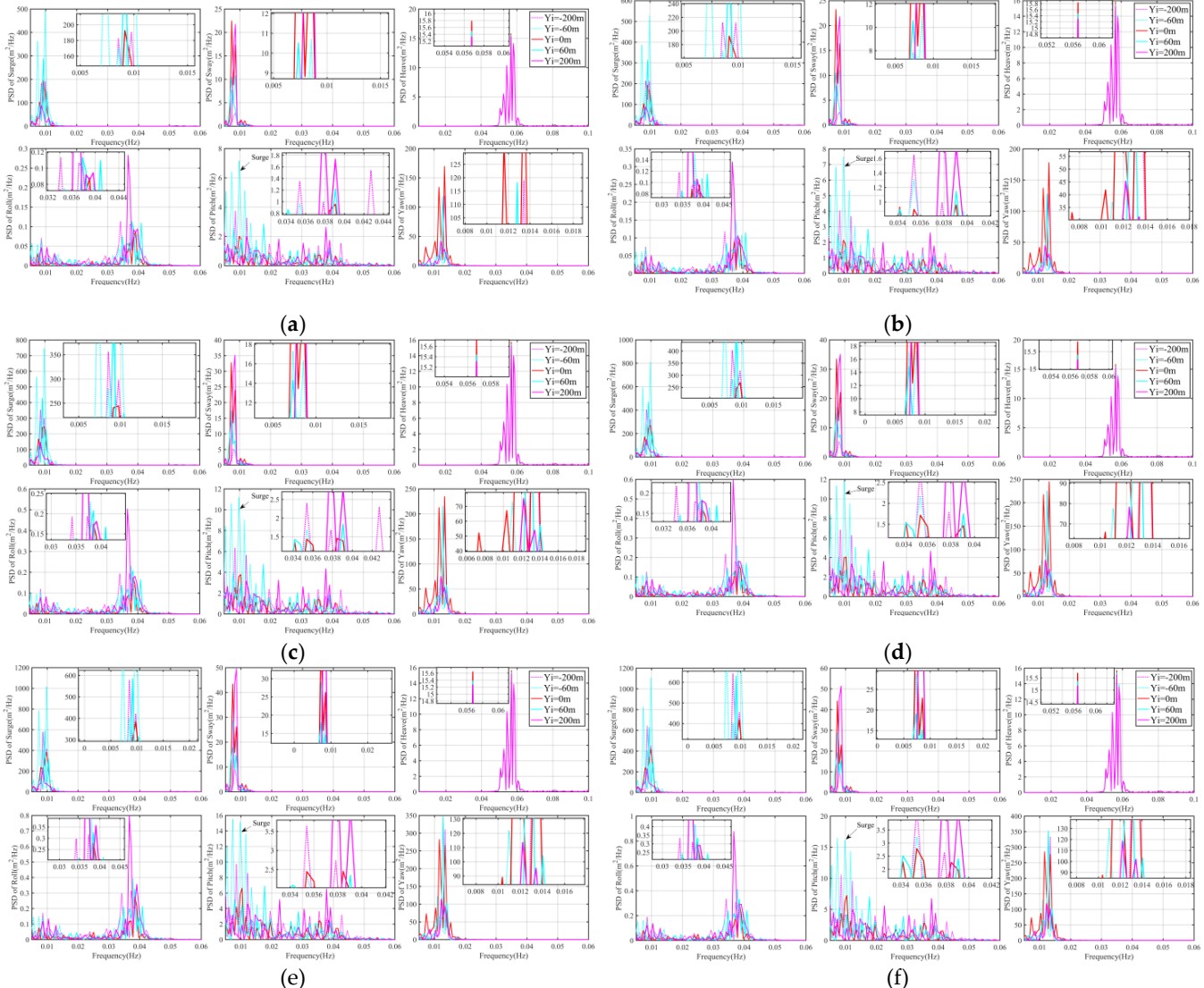

**Figure 18.** Low-frequency platform motions response of FOWT 2 under particular wake inflow conditions: (**a**) 8_0.06_0.13, 5D streamwise spacing; (**b**) 8_0.06_0.17, 5D streamwise spacing; (**c**) 8_0.08_0.13, 5D streamwise spacing; (**d**) 8_0.08_0.17, 5D streamwise spacing; (**e**) 8_0.10_0.13, 5D streamwise spacing; (**f**) 8_0.10_0.17, 5D streamwise spacing.

### 3.4. Structural Loading

This study focused on the analysis of how the structural loading behaved for a waked FOWT (FOWT 2) under different wake inflow conditions. Fatigue including out-of-plane moment at the blade root, and tower base fore-aft moment, was examined.

### 3.4.1. Damage Equivalent Loads Analysis

Damage equivalent loads under simulation cases mentioned above were calculated to compare the effects of wake interference on load and dynamic performance of FOWT. Damage equivalent loads (DEL) were calculated according to Equation (4):

$$DEL = \left( \sum_{i=1}^{n} \frac{S_i^m}{N_{eq}} \right)^{\frac{1}{m}} \tag{4}$$

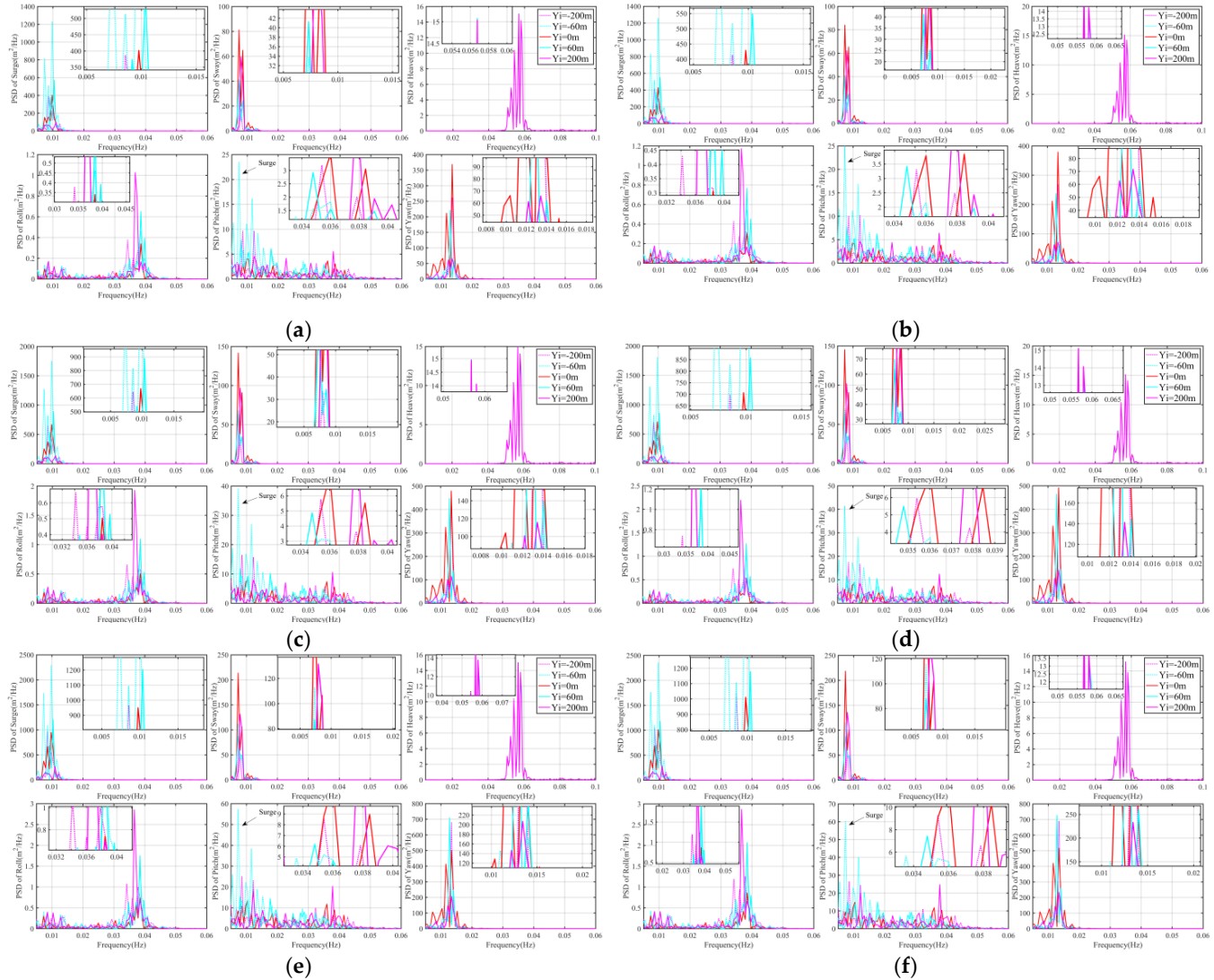

**Figure 19.** Low-frequency platform motions response of FOWT 2 under particular wake inflow conditions: (**a**) 10_0.06_0.13, 5D streamwise spacing; (**b**) 10_0.06_0.17, 5D streamwise spacing; (**c**) 10_0.08_0.13, 5D streamwise spacing; (**d**) 10_0.08_0.17, 5D streamwise spacing; (**e**) 10_0.10_0.13, 5D streamwise spacing; (**f**) 10_0.10_0.17, 5D streamwise spacing.

The Wöhler exponents were defined as m = 10 for composite materials (blades) and m = 4 in case of steel (tower). The DEL was not corrected for mean load effects [43].

In this subsection, normalized DEL was defined as follows:

$$DEL\_{Normalized} = \frac{DEL\_j}{DEL\_standard} \tag{5}$$

where $DEL\_j$ were the DELs calculated by Equation (4), and $DEL\_standard$ were the DELs of FOWT 2 at Yi = −200 m with different wind conditions and streamwise spacing, the details were shown in Figures 21–23.

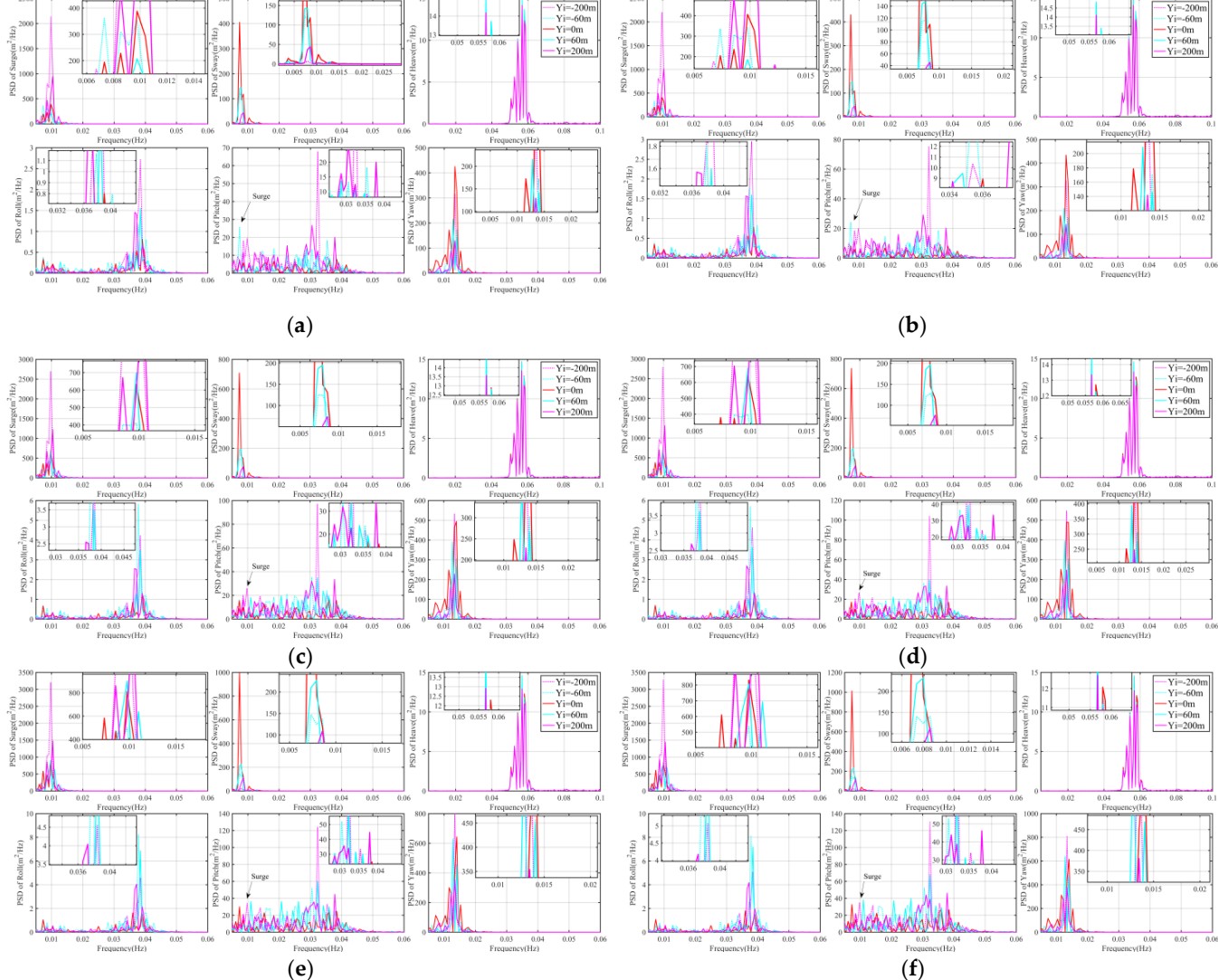

**Figure 20.** Low-frequency platform motions response of FOWT 2 under particular wake inflow conditions: (**a**) 12_0.06_0.13, 5D streamwise spacing; (**b**) 12_0.06_0.17, 5D streamwise spacing; (**c**) 12_0.08_0.13, 5D streamwise spacing; (**d**) 12_0.08_0.17, 5D streamwise spacing; (**e**) 12_0.10_0.13, 5D streamwise spacing; (**f**) 12_0.10_0.17, 5D streamwise spacing.

Figure 21a showed that the DELs at the blade root have a great increase from right-half wake to no wake (from Yi = −60 m to Yi = −130 m). The same trend can be found in Figures 22a and 23a. Figures 21c and 22c clearly demonstrated that the DELs at tower base display an incomplete symmetry, taking the full wake (Yi = 0 m) as the center. Further, the DELs on the right side of symmetry center were larger than DELs on the right side of symmetry center.

The increase in TI led to a significant amplification in the DELs at the blade root as well as the DELs at tower base. Amplification of the DELs at tower base due to increase in wind shear were not as obvious as amplification of the DELs at the blade root.

Maximum amplification of the DELs at the blade root with 5D streamwise spacing ranged from 0.35 to 0.2, while it changed from 0.33 to 0.17 with 10D streamwise spacing, presented in Figures 21a,b, 22a,b and 23a,b. Maximum amplification of the DELs at tower base with 5D streamwise spacing ranged from 0.31 to 0.20, while it changed from 0.22 to 0.16 with 10D streamwise spacing, demonstrated in Figures 21c,d, 22c,d and 23c,d.

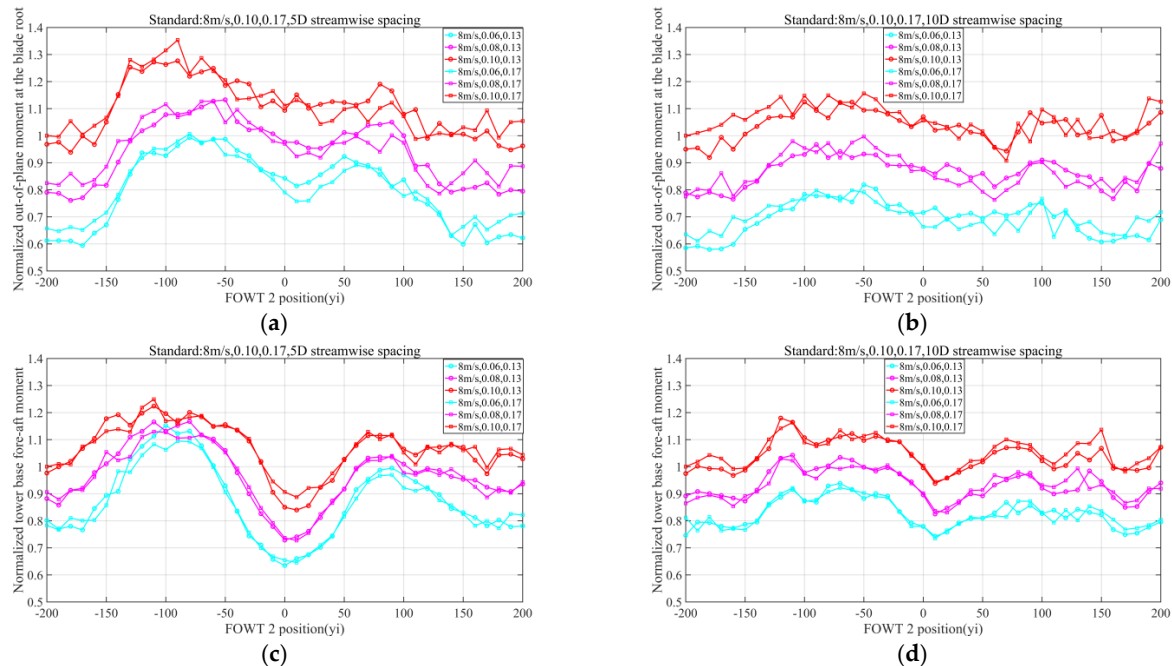

**Figure 21.** Normalized damage equivalent loads of FOWT 2 under different wake inflow conditions: (**a**) out-of-plane moment at the blade root, 8 m/s, 5D streamwise spacing; (**b**) out-of-plane moment at the blade root, 8 m/s, 10D streamwise spacing; (**c**) tower base fore-aft moment, 8 m/s, 5D streamwise spacing; (**d**) tower base fore-aft moment, 8 m/s, 10D streamwise spacing.

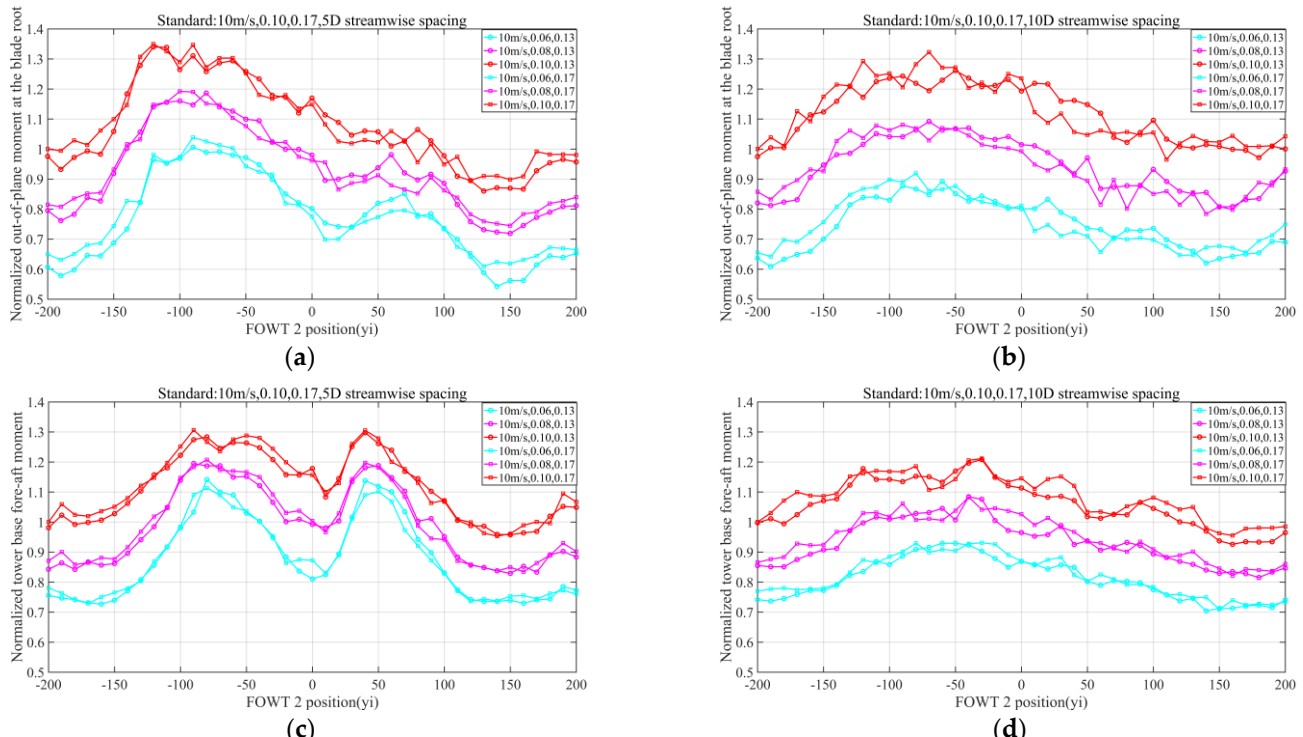

**Figure 22.** Normalized damage equivalent loads of FOWT 2 under different wake inflow conditions: (**a**) out-of-plane moment at the blade root, 10 m/s, 5D streamwise spacing; (**b**) out-of-plane moment at the blade root, 10 m/s, 10D streamwise spacing; (**c**) tower base fore-aft moment, 10 m/s, 5D streamwise spacing; (**d**) tower base fore-aft moment, 10 m/s, 10D streamwise spacing.

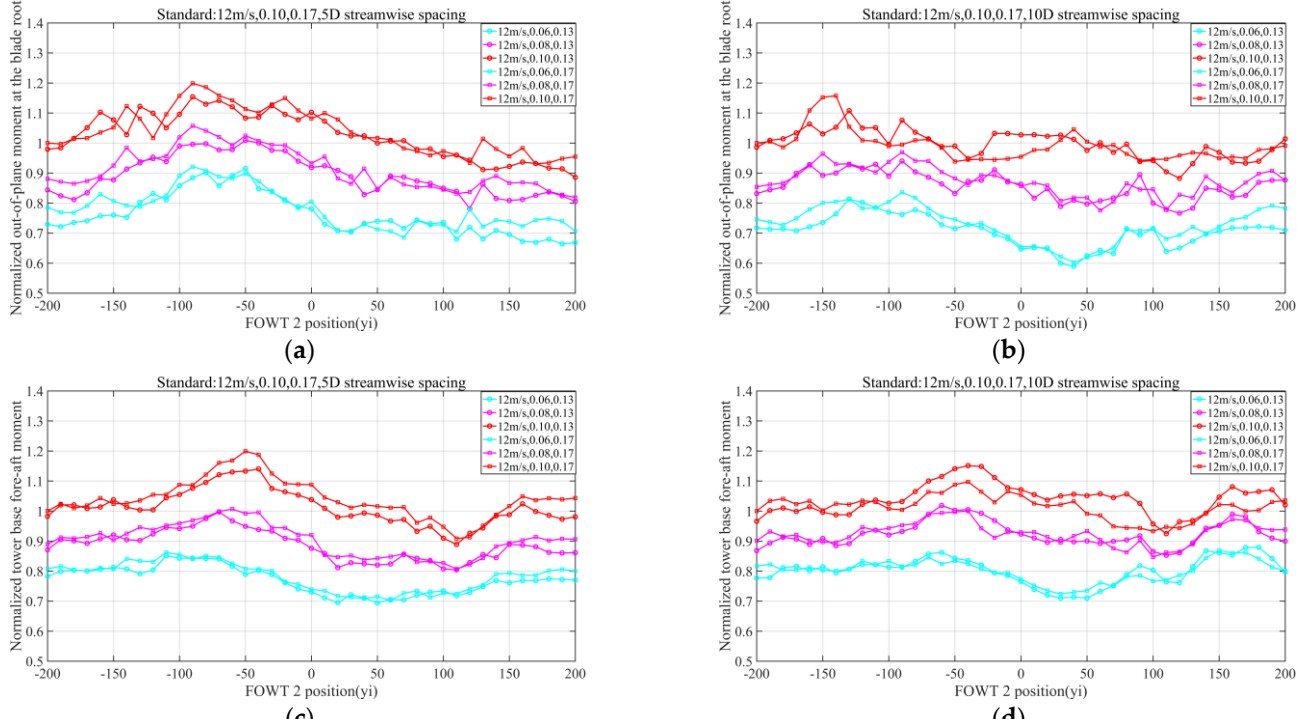

**Figure 23.** Normalized damage equivalent loads of FOWT 2 under different wake inflow conditions: (**a**) out-of-plane moment at the blade root, 12 m/s, 5D streamwise spacing; (**b**) out-of-plane moment at the blade root, 12 m/s, 10D streamwise spacing; (**c**) tower base fore-aft moment, 10 m/s, 5D streamwise spacing; (**d**) tower base fore-aft moment, 12 m/s, 10D streamwise spacing.

### 3.4.2. Power Spectrum Analysis

Since the FOWT 2s were subject to identical wave excitations in the turbulent wind, the amplitude of structural loads could be purely attributed to the wind force. In the interest of conciseness, the following figures had presented only the result of mean wind speed at 8 m/s in this subsection.

Figure 24 illustrated the response character of out-of-plane moment at the blade root. The majority of the response energy were located within the low-frequency range, mainly induced by the TI. Besides, the response was also observed around 0.28 Hz, namely the rotor frequency (2P). Although previous work has found that wake increases load responses at higher frequencies, such as the rotor frequency (1P) and blade-passing frequency (3P) [41], the phenomenon was focused on for the first time. The 2P frequency response was induced by the spatial inhomogeneity of the wind field, as a result of asymmetrical wake inflow, wind shear and the phase lag between two points in the rotor plane.

Figure 25 demonstrated the significant response of tower base fore-aft moment. Although the 1P frequency response (around 0.28 Hz) was not sensitive to wake, the 3P frequency response (around 0.42 Hz) was amplified by wake, especially for asymmetrical wake (Yi = −60 m and Yi = 60 m). Moreover, in some situations, half wake led to more than twice the resonant response as full wake.

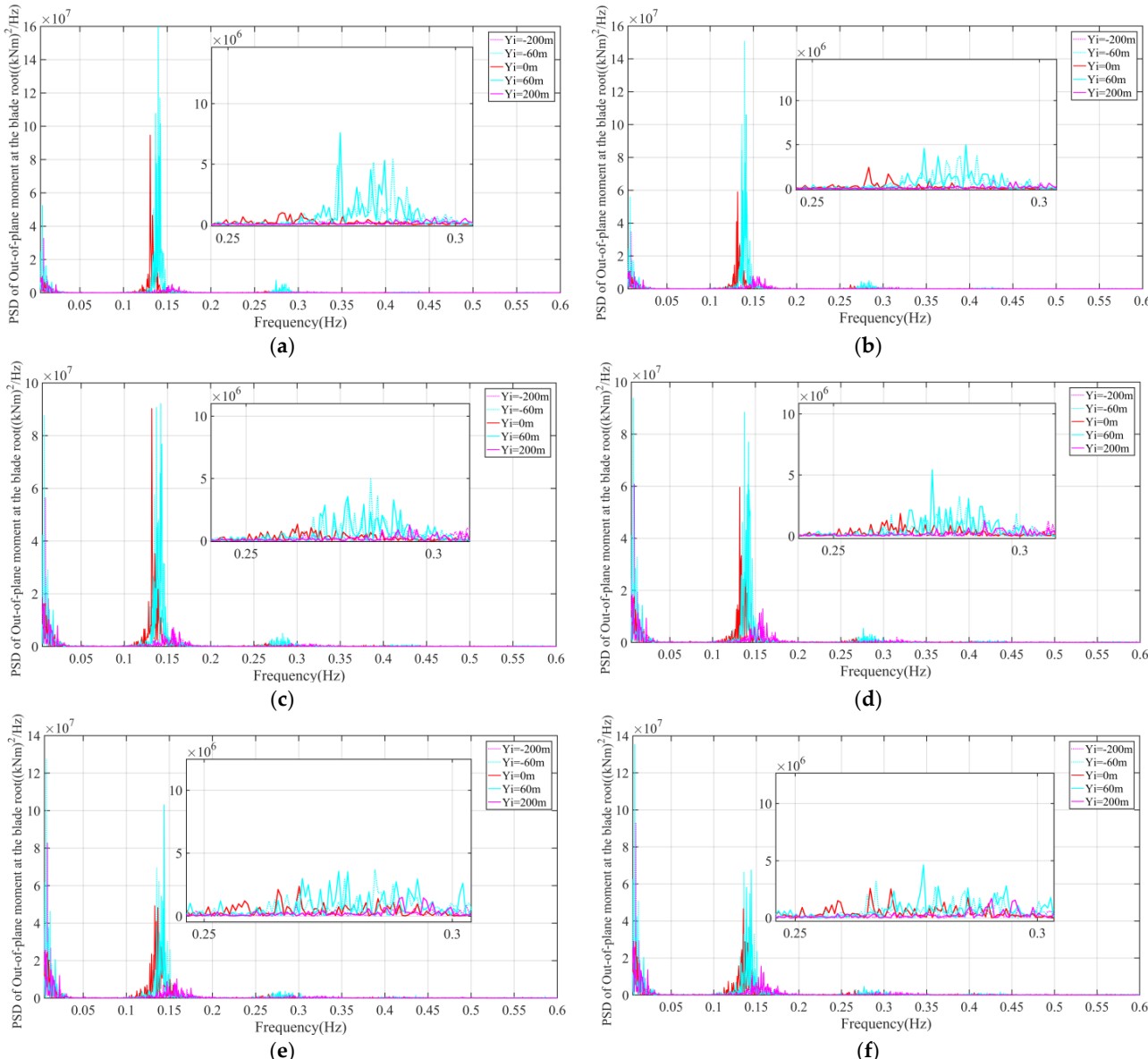

**Figure 24.** PSD of out-of-plane moment at the blade root of FOWT 2 under particular wake inflow conditions: (**a**) 8_0.06_0.13, 5D streamwise spacing; (**b**) 8_0.06_0.17, 5D streamwise spacing; (**c**) 8_0.08_0.13, 5D streamwise spacing; (**d**) 8_0.08_0.17, 5D streamwise spacing; (**e**) 8_0.10_0.13, 5D streamwise spacing; (**f**) 8_0.10_0.17, 5D streamwise spacing.

The comparison in the Figures 24 and 25 further illustrated the correctness of the hypothesis. Specifically, the frequency response of the FOWT increased most obviously when the right half rotor was disturbed by the wake.

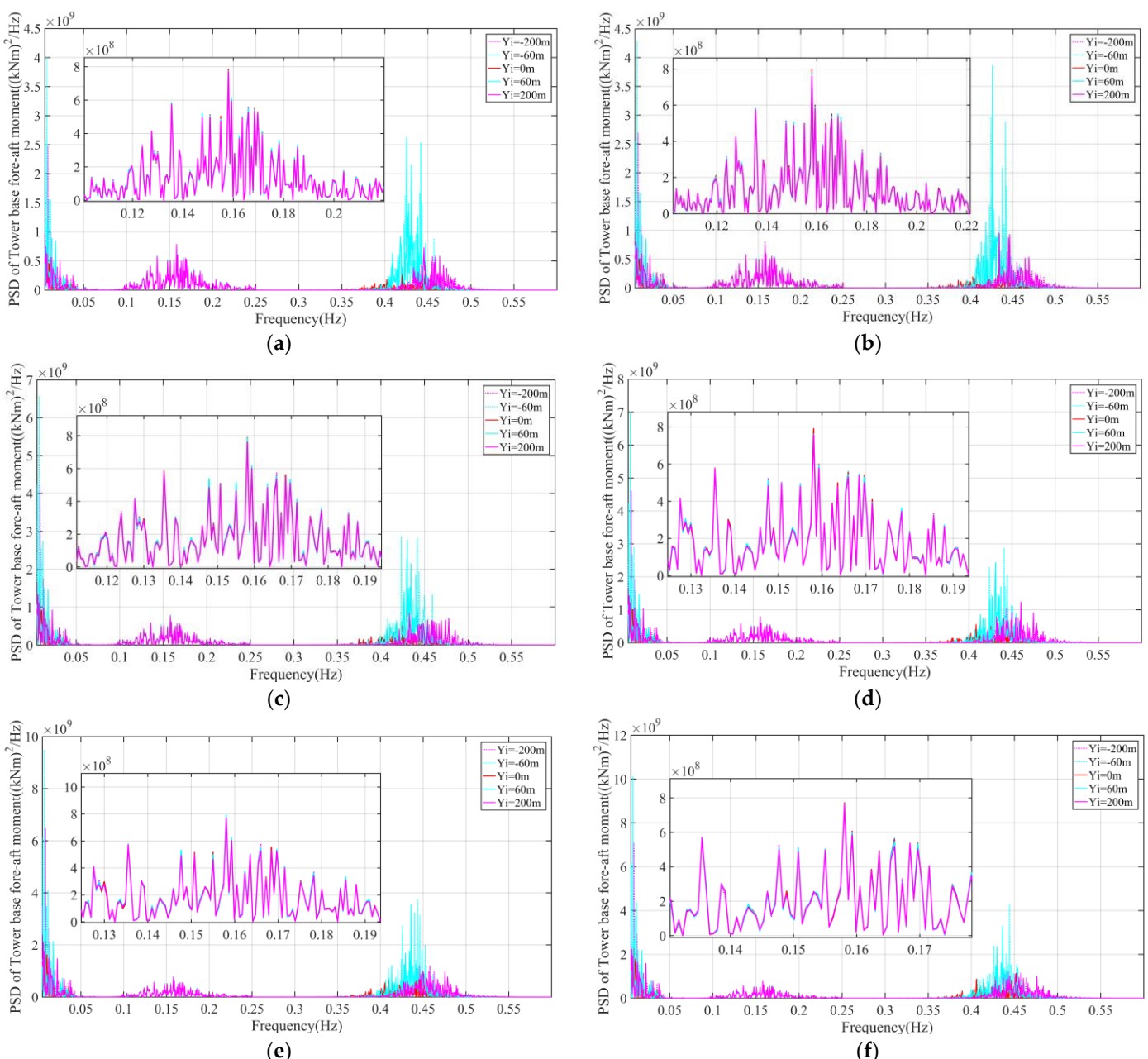

**Figure 25.** PSD of tower base fore-aft moment of FOWT 2 under particular wake inflow conditions: (**a**) 8_0.06_0.13, 5D streamwise spacing; (**b**) 8_0.06_0.17, 5D streamwise spacing; (**c**) 8_0.08_0.13, 5D streamwise spacing; (**d**) 8_0.08_0.17, 5D streamwise spacing; (**e**) 8_0.10_0.13, 5D streamwise spacing; (**f**) 8_0.10_0.17, 5D streamwise spacing.

## 4. Discussion and Conclusions

The present investigation concentrates on two tandem semisubmersible FOWTs under different wake inflow conditions, in the consideration of streamwise spacing, TI and wind shear. By controlling for variables, the effect of each factor on FOWT is clearly observed.

### 4.1. Discussion

The power loss of FOWT under the full wake can be 60%, 55% and 30% when the mean wind speed was 8 m/s, 10 m/s and 12 m/s, respectively, even though 5D was away from the former FOWT. The power increased with almost the same trend from full wake to no wake at both ends. FOWTs under right-half wake did not perform as well as FOWTs

whose rotor was under left wake. Moreover, increasing the distance effectively improves the power recovery, especially when it was close to the rated wind speed. In addition, the effect of wind shear on wake recovery was less than that of TI. The wake not only caused the power loss of FOWT 2, but also increased the power fluctuation significantly which cannot be ignored for connecting wind farm to grid, especially when rotor of FOWT 2 under right-half wake.

A crucial part of the investigation on platform motion is to evaluate the effects of wake interactions. Different wake inflow conditions caused evident inequable fluctuations on platform surge and pitch motion, which means that the stability of FOWT is threatened. Through the comparative analysis of different factors, it is found that the semi-wake under high TI is a great threat to the platform surge and pitch motion, and this is related to the low frequency response caused by high turbulence intensity incoming wind, as shown in Figure 17a.

DEL analysis and power spectrum analysis further investigated the response of FOWT in the wake condition. In contrast, half wake led to obvious elevation on the resonant response in some situations, and this phenomenon was in line with [44]. These results were within the hypothesis and illustrate the problem in more detail. Many previous studies have found that the wake effects lead to increased fatigue loading for onshore or bottom-fixed offshore WTs. In the present work, we find that for FWTs, the influence of wake flow on structural loading is more obvious in the case of high TI and high wind shear for the below rated wind speed scenarios.

### 4.2. Conclusions

In this paper, FAST.Farm was used for testing FOWTs under different wake inflow conditions, exploiting the characteristics of the semi-submersible FOWT influenced by wake turbine interference effects.

The conclusions can be summarized as follows:

(1) For the analysis of power performance, we focus on factors affecting wake recovery and power fluctuation. Moreover, the increase in power fluctuations due to different wake inflow conditions will not only increase the difficulty of wind power forecasting, but also increase the cost of wind farm operation, such as adding battery energy storage systems.

(2) Platform motions of FOWT 2 became increasingly violent under different wake inflow conditions, especially surge and pitch in half wake, which was clearly indicated through both time and frequency domain analysis.

(3) Structural loading of a waked FOWT were affected seriously by different wake inflow conditions and TI. In contrast, structural loading was not sensitive to the wind shear. Moreover, a great increase in DELs and a significant resonant response indicated that the impact of half wake on the FOWT was more serious than that of full wake.

(4) FOWTs whose rotor under right-half wake inflow condition suffered the most severe wake, taking the above factors into consideration.

### 4.3. Limitations of the Study

In this paper, the offset of wake center was not taken into account when defining the wake inflow condition. Moreover, the simulation results need to be compared and verified with the measured data of the actual wind farm.

### 4.4. Future Perspective on the Research

According to the simulation research in this paper, the health status and maintenance cost of FOWT, as well as reducing the time when the FOWT operates under half wake condition, should be comprehensively considered during the design of offshore wind farm. In addition, by trying to prevent half wake condition, more comprehensive active wake control strategies can be scrutinized, finding a balance between increasing power and reducing structural loading.

**Author Contributions:** Conceptualization, L.X. and Y.X.; methodology, L.X.; formal analysis, L.X.; data curation, Y.X.; writing—original draft preparation, L.X.; writing—review and editing, L.X., J.W., L.Z. and Y.X.; visualization, L.X., Z.W. and M.Y.; funding acquisition, Y.X. All authors have read and agreed to the published version of the manuscript.

**Funding:** This research was funded by Shandong Provincial Natural Science Foundation, grant number ZR2021ZD23. This research was also funded by Offshore Wind Power Intelligent Measurement and Control Research Centre and Laboratory Construction at the Ocean University of China, grant number 861901013159.

**Institutional Review Board Statement:** Not applicable.

**Informed Consent Statement:** Not applicable.

**Data Availability Statement:** Not applicable.

**Acknowledgments:** We wish to thank Jonkman Jason of National Renewable Energy Laboratory and Yang Yang of Ningbo University.

**Conflicts of Interest:** The authors declare that they have no known competing financial interest or personal relationships that could have appeared to influence the work reported in this paper.

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
