# Peer review of "Wake Interactions of Two Tandem Semisubmersible Floating Offshore Wind Turbines Based on FAST.Farm"

_jmse, doi:10.3390/jmse10121962_

Round 1
Reviewer 1 Report
Dear authors,
The topic of your manuscript is interesting and relevant. However, I believe the manuscript needs to be significantly improved before it can be ready for publication. I have the following comments,
The structure and narrative of the manuscript is not clear. The introduction is adequate but the following sections do not have a clear narrative and topics were introduced without a clear sequence. I suggest entirely restructuring the manuscript into the traditional sections of a research paper: introduction, methods & materials, results, discussion and conclusion.
The scientific contribution and novelty of the research are not clear. I highly recommend to explicitly indicate the hypothesis of your research and afterwards in the results section indicate if the hypothesis was proven or disproven by the results generated by the research.
The methods are not clear. I suggest to restructure this section and add a flow chart to illustrate to the reader the application of the method. It will be important to indicate the advantages of using the proposed method versus other alternatives mentioned in the introduction.
There are several figures and tables in the method section that appear to be results. This needs to be restructured.
The results section is highly confusing, without a clear narrative. Many figures and tables are stacked one after the other without an individual description or discussion preceding each one. This needs to be changed. Every figure, chart or figure needs to have their own description or discussion, highlighting how this leads to the scientific contribution of the research and to prove the research hypothesis.
For instance, Figure 1-8 and tables 1 – 4 are stacked one after the other without any description before each of them. This significantly hinders the narrative of the manuscript and prevent readers from evaluating the scientific contribution.
There is a section called Discussion and Conclusion. However, there is no discussion at all. There should be an independent Discussion section that highlights the relevance of the research and its results.
The conclusion needs to be rewritten. It contains discussion instead of conclusion. The limitations of the study need to be expanded because at this point only one factor is considered. Selecting a particular method to perform research generates a number of limitations that need to be disclosed to the reader.
Future research is too abstract. Mentioning that this research it will inspire other researchers, without indicating the future path of the research does not provide clarity to the readers.
Author Response
Dear editor and reviewer,
Thank you very much for your comments and suggestions.
Those comments are all valuable and very helpful for revising and improving our paper, as well as the important guiding significance to our researches. We have studied comments carefully and have made correction which we hope meet with approval. Revised portion are marked in red in the paper. The main corrections in the paper and the responds to the reviewer’s comments are as flowing:

Reviewer 2 Report
This study aims to investigate the wake interactions of offshore wind farms. This work is well investigated and presented. However, before getting it published, there are some drawbacks need to get addressed first:
· I suggest that the authors add some details on the turbulence model (IECKAI), utilized in their simulation, in the methodology section.
· For more clarification, please replace larger pictures for Figure 1. You can put them vertically as well.
· The percentage sign (%) must be placed close to the number, with no space between.
· The amount of text is low regarding elaborations on figures 5-7 and figures 16-18; please add more description.
· To improve the quality of the introduction section, the following article is suggested:
“Wind Farm Layout Optimization with Different Hub Heights in Manjil Wind Farm Using Particle Swarm Optimization” https://doi.org/10.3390/app11209746

Author Response

(The authors gave the same response as above.)

Round 2
Reviewer 1 Report
Dear authors,
Most of the review comments were addressed in the revised version of your manuscript. I have the following comments for the revised version of your manuscript.
I reiterate my suggestion that in the results section it is indicated if the hypothesis was proven or disproven by the results generated by the research.
I suggest improving Figure 1 for the flowchart. It is blurry and some of the font size is too small to read clearly.
I believe that adding additional description or discussion before the figure, chart or figure would help readers better understand its relevance and how it proves or disproves the research hypothesis.
The font size of the axis labels, other labels and legends for most of the figures is too small and difficult to read. I suggest improving all the figures.